# Converging and evolving immuno-genomic routes toward immune escape in breast cancer

Juan Blanco-Heredia [1,2,3,14], Carla Anjos Souza [1,2,14], Juan L. Trincado [4,5,14], Maria Gonzalez-Cao [6], Samuel Gonçalves-Ribeiro [7], Sara Ruiz Gil[4], Dmytro Pravdyvets[8], Samandhy Cedeño [1], Maurizio Callari [9], Antonio Marra[3], Andrea M. Gazzo[3], Britta Weigelt [3], Fresia Pareja [3], Theodore Vougiouklakis[3], Achim A. Jungbluth[3], Rafael Rosell [6], Christian Brander [1,10,11], Francesc Tresserra[6], Jorge S. Reis-Filho [3], Daniel Guimarães Tiezzi [12,13,14], Nuria de la Iglesia [1], Holger Heyn [4,8] & Leticia De Mattos-Arruda [1,2] ✉

The interactions between tumor and immune cells along the course of breast cancer progression remain largely unknown. Here, we extensively characterize multiple sequential and parallel multiregion tumor and blood specimens of an index patient and a cohort of metastatic triple-negative breast cancers. We demonstrate that a continuous increase in tumor genomic heterogeneity and distinct molecular clocks correlated with resistance to treatment, eventually allowing tumors to escape from immune control. TCR repertoire loses diversity over time, leading to convergent evolution as breast cancer progresses. Although mixed populations of effector memory and cytotoxic single T cells coexist in the peripheral blood, defects in the antigen presentation machinery coupled with subdued T cell recruitment into metastases are observed, indicating a potent immune avoidance microenvironment not compatible with an effective antitumor response in lethal metastatic disease. Our results demonstrate that the immune responses against cancer are not static, but rather follow dynamic processes that match cancer genomic progression, illustrating the complex nature of tumor and immune cell interactions.

Breast cancers are complex dynamic systems that arise in the context of spatially structured genomic and immune microenvironments[1]. The relative abundance of somatic mutations suggesting the presence of neoantigens capable of activating specific T cells is one of the key characteristics that make triple-negative breast cancer (TNBC) more likely to respond to immunotherapy than other breast cancer subtypes[2].

Previous studies have demonstrated that immunosuppressive interactions between tumor cells, surrounding stromal, and immune cells might support metastatic progression and escape from immune control, challenging the efficacy of cancer immunotherapy[3–5]. Some of

the major immune escape mechanisms that maximize the probability of a tumor to progress include loss of neoantigen presentation, immune cell exhaustion, and the presence of an immunosuppressive microenvironment[6]. At the genomic level, mutation rate, genomic instability, and pool size are the drivers of diversity, which may constitute driving forces imposed by immune pressures and consequently may lead to differential responses to therapies[7–9]. However, whether TNBC cancers diversify over time and across spatially distinct synchronous metastatic areas, evolving along a molecular clock into genomically diverse landscapes with distinct immune environments has yet to be understood.

To explore the sequential and parallel interplay between anti-tumor immunity and TNBC metastases, leading to immune escape, we extensively characterize a TNBC index patient leveraging single-cell peripheral blood T cell RNA and T cell receptor (TCR) sequencing, tumor bulk DNA, RNA and TCR sequencing, neoantigen T cell reactivity, clinical and immunohistologic data. We then extend the analyses to a series of eleven primary TNBCs and postmortem metastases previously published by our group and others[10,11]. Here, we identify the routes by which immune escape is driving lethal disease and show that the interplay between TNBC metastasic cells and host antitumor immunity determines co-existing mechanisms of immune escape within the same patient, with potential implications for combinatory immunotherapies and biomarker development.

## Results

### Clinical and sample characteristics of TNBC patients

Overall, 112 specimens from 12 metastatic TNBC breast cancer patients were available, consisting of 11 primary tumors, 15 sequential on-treatment metastasis (over time), 18 serial blood samples, 2 body fluids (ascitic and pleural), as well as 66 parallel multiregion metastases (either synchronous, affecting the same metastatic site or metastases in different anatomical sites within the same patient) from warm autopsies (Fig. 1, Supplementary Fig. 1). An index patient with TNBC was closely monitored from diagnosis through metastatic progression for 2033 days until she expired. Data included samples from various stages of the disease, such as a primary tumor specimen, 15 sequential chest wall skin biopsies during treatment, 18 blood samples, 2 body

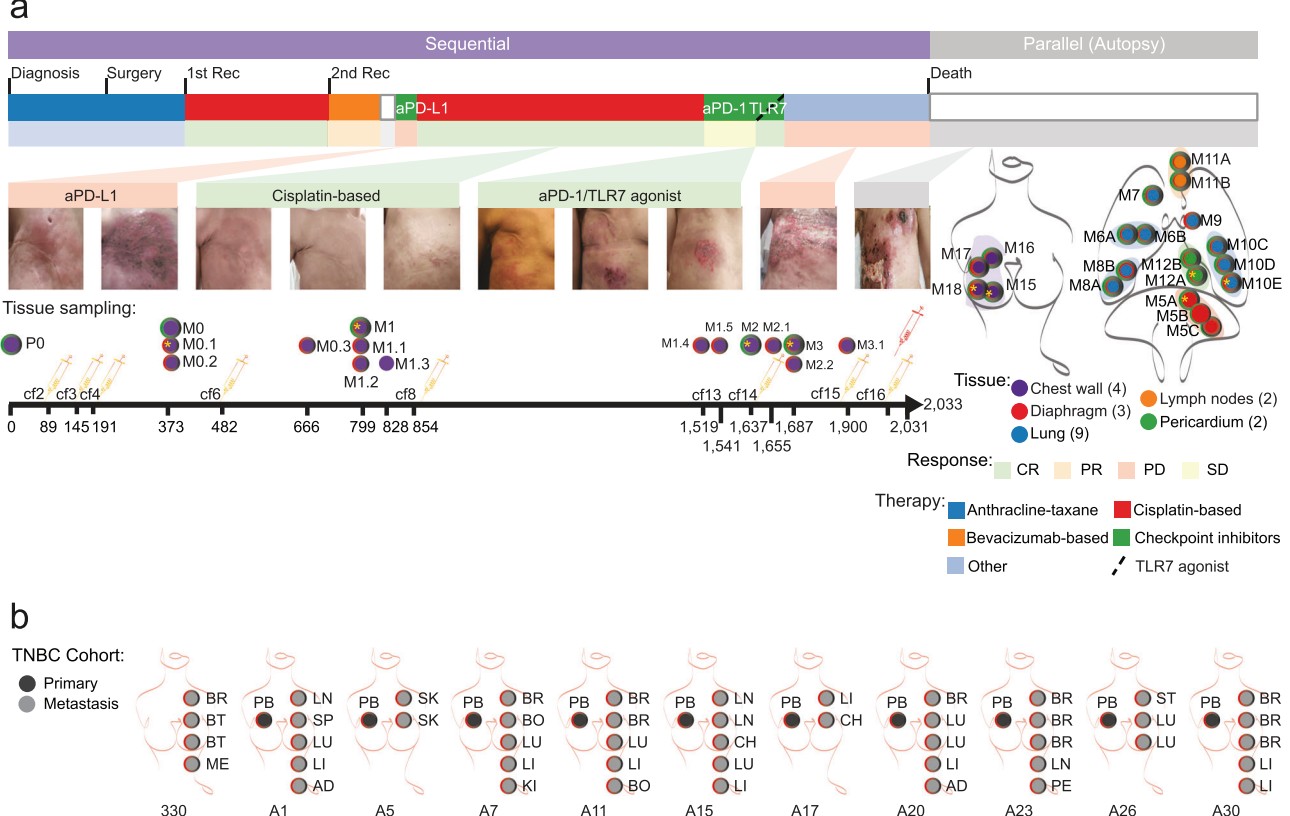

**Fig. 1 | Study schematics. a** Schematics of the study and anatomical map of biospecimen collection for sequential and parallel multiregion analyses of the index TNBC patient. Timeline is provided in days from the diagnosis. The index patient presented here was a 49-year-old woman diagnosed with a stage III TNBC (T2N3, estrogen receptor (ER), progesterone receptor (PR) and HER2 0+ negative, Ki67 60%, grade 3) with a 3.5 cm right breast cancer mass and lymph node involvement, who underwent multiple systemic therapies due to recurrences and metastatic progression over 2033 days of clinical follow-up. She underwent neoadjuvant chemotherapy with anthracycline and taxane achieving a pathological complete response after mastectomy. The patient presented multiple clinical recurrences at the chest wall from day 373, and achieved complete response with cisplatin-based therapy, surgery, and local radiotherapy. A second chest wall recurrence occurred around day 666 with partial response to bevacizumab-based therapy. Immunotherapy employing anti-PD-L1 monoclonal antibody atezolizumab was administered on day 799 after diagnosis, followed by in a rapid disease progression. However, a long-lasting complete response of 22 months was evidenced after a re-challenge of cisplatin and gemcitabine (day-854 to day 1519), after a previous response to the same drug had been 2.2 fold shorter (day 373 to day 666). The anti-PD-L1 administration before the re-challenge with cisplatin, although culminating in a rapid disease progression, could have contributed to the subsequent long-lasting antitumor response to cisplatin, motivating our investigation of immune escape. Subsequently, the patient presented a progression at the chest

wall, and received anti-PD-1 (pembrolizumab) and chemotherapy, with stable disease for 4 months. Then, upon chest wall progression, pembrolizumab plus toll-like receptor (TLR) 7 agonist (topical) was administered in the chest wall metastases with a transient local complete response that lasted around 50 days. The patient received other lines of systemic therapy (i.e. palbociclib followed by cyclophosphamide, pegylated liposomal doxorubicin, cisplatin plus gemcitabine, paclitaxel plus bevacizumab, eribulin) (Supplementary Table 1) and expired on day 2033. Sequential chest wall images illustrate the clinical evolution of a TNBC patient over time. Postmortem parallel multiregion metastases were synchronous, affecting the same metastasis or metastases affecting different anatomical sites (separated into 2 or 3 sites when indicated) within the index patient as indicated. aPD-L1 anti-programmed death-ligand 1 monoclonal antibody, aPD-1 anti-programmed cell death protein 1 monoclonal antibody, TNBC triple-negative breast cancer, CR complete response, cf circulating free (DNA), M metastasis, P primary tumor, PR partial response, PD progressive disease, Rec recurrence, SD stable disease, TLR7 Toll-like receptor 7. **b** Schematics of 11 TNBC patients cohort included in the study as validation cohort and subjected to re-analysis of their WES and bulk RNA-seq data for primary tumors (n = 10) and parallel multiregion metastases (n = 46). AD adrenal, BO bone, BR brain, BT breast [metastasis], CH chest, KI kidney, LN lymph node, LU lung, LI liver, ME meninges, PB primary breast, PE pleura, SP Spine, ST soft tissue, SK skin.

fluids (ascitic and pleural), and 20 parallel metastases from multiple regions at autopsy.

The patient, a 49-year-old woman, initially had stage III TNBC with a 3.5 cm breast mass and lymph node involvement. She underwent various treatments, achieving a complete response after neoadjuvant chemotherapy. Recurrences occurred over time, and immunotherapy with atezolizumab showed a fast disease progression but later contributed to a prolonged response to cisplatin and gemcitabine. The patient underwent several treatment regimens for metastatic disease, including cytotoxic and immunotherapies, and ultimately passed away after 2033 days of clinical follow-up (Fig. 1, Supplementary Table 1 and Supplementary Data 1).

As an expansion cohort, a series of 11 TNBC patients with bulk DNA-seq and RNA-seq from 10 primary breast cancers and 46 parallel multiregion metastases (e.g., lymph nodes, breast, CNS, adrenal, liver, lung-pleural, bone, kidney, soft tissue, skin) sampled with warm autopsy procedures were included from data previously published by our group and others[10,11] (Supplementary Data 1).

## TCR repertoires lose diversity over time leading to convergent evolution as breast cancer progresses

T cells are key mediators of adaptive antitumor immunity, but how T cells function and the TCR repertoire is shaped over time during cancer progression remains poorly understood[12]. To interrogate the evolution of the T cell repertoire from the TNBC index patient and to identify patterns of immune evasion, we performed two complementary approaches. Firstly, we profiled the TCR beta chain (TCRβ) of genomic DNA from bulk tumor tissue of the TNBC index patient at different time points: 4 on-treatment sequential chest wall (M0-M3) and 5 parallel multiregion metastases sampled at autopsy (Fig. 2a). Secondly, we carried out single-cell RNA-sequencing (scRNA-seq) and T cell receptor sequencing (scTCR-seq) of peripheral blood T cells collected 48 h before the TNBC index patient's death and characterized their immunophenotypes of different T cell populations and clonotypes (expanded T cells). Finally, we integrated the TCR identities to link immunophenotype, tissue, and clonotypes information to track TCR repertoire evolution (Fig. 2a).

Our TCRβ deep sequencing generated 72,831 templates for all our samples (3713 SD) and a range of 1455–5091 (1147 SD) unique, productive TCRβs per sample. A total of 21,164 unique amino acid sequences from all nine samples were used to calculate (i) the proportion of *private* TCR sequences in on-treatment sequential chest wall metastases ($N = 11,876$), (ii) *private* TCR sequences in parallel multiregion metastases ($N = 7715$), or (iii) shared TCR sequences ($N = 1573$) (Supplementary Fig. 2a).

Next, we performed a TCR network analysis using Levenshtein distance, and used the distance of 1 amino acid change as the threshold of similarity to establish edges or connections in the network (see Methods). About 4185 TCR sequences were connected, and of these, 2167 and 1642 were present privately in the sequential and parallel multiregion metastases of the TNBC index patient, respectively.

We observed that the TCR repertoire partially overlapped between the sequential chest wall metastases and parallel multiregion metastases of the TNBC index patient, with the later time point metastases presenting more shared TCR sequences (Fig. 2b). This indicates that the TCR repertoire is highly dynamic, changing and evolving along the clinical course of the disease, with metastases closer in time, sharing more TCR sequences than metastases collected more apart in time. The M3-day 1687 time point (tumor locally treated with TLR7 agonist) showed convolution of T cell repertoires with a greater TCR network connectivity than the previous time points, based on metrics such as density, average clustering coefficient and S-metric (Supplementary Fig. 2b). Average clustering coefficient and S-metric's values increased in later time points in longitudinal metastases and when integrated with postmortem parallel multiregion metastases.

The M15 chest wall sample at autopsy resembled the longitudinal chest wall metastases, while M18 (spatially different area of the chest wall also sampled at autopsy) showed a distinct TCR repertoire compared to the other synchronous metastases. This may suggest that the immunogenic neoantigens eliciting an adaptive T cell response may be different between M15 and M18, which may have evolved along separate genomic evolution paths. To validate these findings, we performed bootstrap analysis, confirming the metrics to be stable and robust (Supplementary Fig. 2b).

Single-cell immunophenotyping of 5024 peripheral blood T cells of the TNBC index patient revealed 11 unique T cell subsets clustered into different activation states (Fig. 2b, c). Each cluster was annotated by determining differentially expressed genes (DEG) based on Wilcoxon rank-sum test (See "Methods" section). Using signatures from a scRNA-seq pan-cancer atlas of T cells[13], we detected T cells in different stages of differentiation, ranging from recently activated effector memory cells (cluster 6), expressing high levels of the early activation molecule CD27, to terminally differentiated effector memory cells (cluster 5; Supplementary Fig. 2c). The largest cluster on our data (cluster 1, 26.92% of total T cells) was composed of effector memory cells expressing IL7R memory marker and AP-1 transcription factors JUN and FOS (Supplementary Fig. 2d). Cluster 2 was found to harbor high levels of γδ and low αβ chain recovery expression (Supplementary Fig. 2e), evidencing the presence of γδ T cells in our dataset (26.13%). A separate cluster (cluster 8), similar to clusters 1 and 2, was characterized by an intermediate gene expression profile between the αβ and γδ clusters (Fig. 2c). Naïve T cells expressing the differentiation markers SELL and CCR7 comprised two clusters of CD4+ and CD8+ T cells (cluster 3, 14.39%; cluster 4, 9.92%, respectively), which clustered close to early activated cluster 6 cells. In depth analysis of this cluster 6 allowed us to unveil four additional clusters: CD4+ T cells in an intermediate state between naïve and effector (cluster 6.0), CD4+ Treg (cluster 6.1), CD8+ cytotoxic (cluster 6.2) and CD8+ proliferating T cells (cluster 6.3; Fig. 2c). Cluster 6.3 showed great transcriptional and clonal resemblance to cluster 6.2 (Fig. 2c), suggesting a common origin. A CD4+ T cell population with a cytotoxic phenotype, characterized by high expression of IL7R, KLRB1, granzymes, and TNF alpha and low expression of CCR7 and SELL was also detected (cluster 7, 4.69%; Fig. 2c, d). These results show that the patient was mounting immune responses detectable in the periphery, even though metastatic disease had fully progressed.

Next, we performed a TCRβ CDR3 repertoire overlap analysis between pairwise samples (here using all the above-mentioned sequential and parallel metastases of the TNBC index patient with the peripheral blood single T cells sampled at day 2031). The analysis was based on the Morisita index[13], which estimates the similarities using the number of shared clonotypes and their abundances. We observed that postmortem samples and peripheral blood single-cell samples harbored a more similar repertoire than those derived from sequential tumors, as expected, given that these samples are chronologically close (48 h apart; Supplementary Fig. 2f–h). The majority of expanded clonotypes at all time points (Fig. 2e) were mapped to the CD8 TEM (cluster 1, 1401 cells, 1079 TCR sequences, 27% of the total) and CD4 CTL T cells (cluster 7, 4.69%; Fig. 2f). We observed that clonotypes present early on in the sequential on-treatment samples were still present in peripheral blood (close to the time of death) and were also found in postmortem metastases samples. By contrast, clones private to parallel multiregion metastases at day 2033 were circulating in blood at day 2031 as recently activated or proliferating CD8+ T cells (i.e. clusters 6.2: CD8 early TEM, cluster 6.3: CD8 Prolif, cluster 5: TEMRA CD8 T cells, in red). Analyses of CD4 and CD8 T cells in tissue show their presence across the disease course but enrichment of T cells in the early sequential metastases and a decline in late metastases of the index patient, according to CD8 ($p$-value $= 0.042$) and CD4 ($p$-value $= 0.059$) immunohistochemistry (IHC) analyses. To

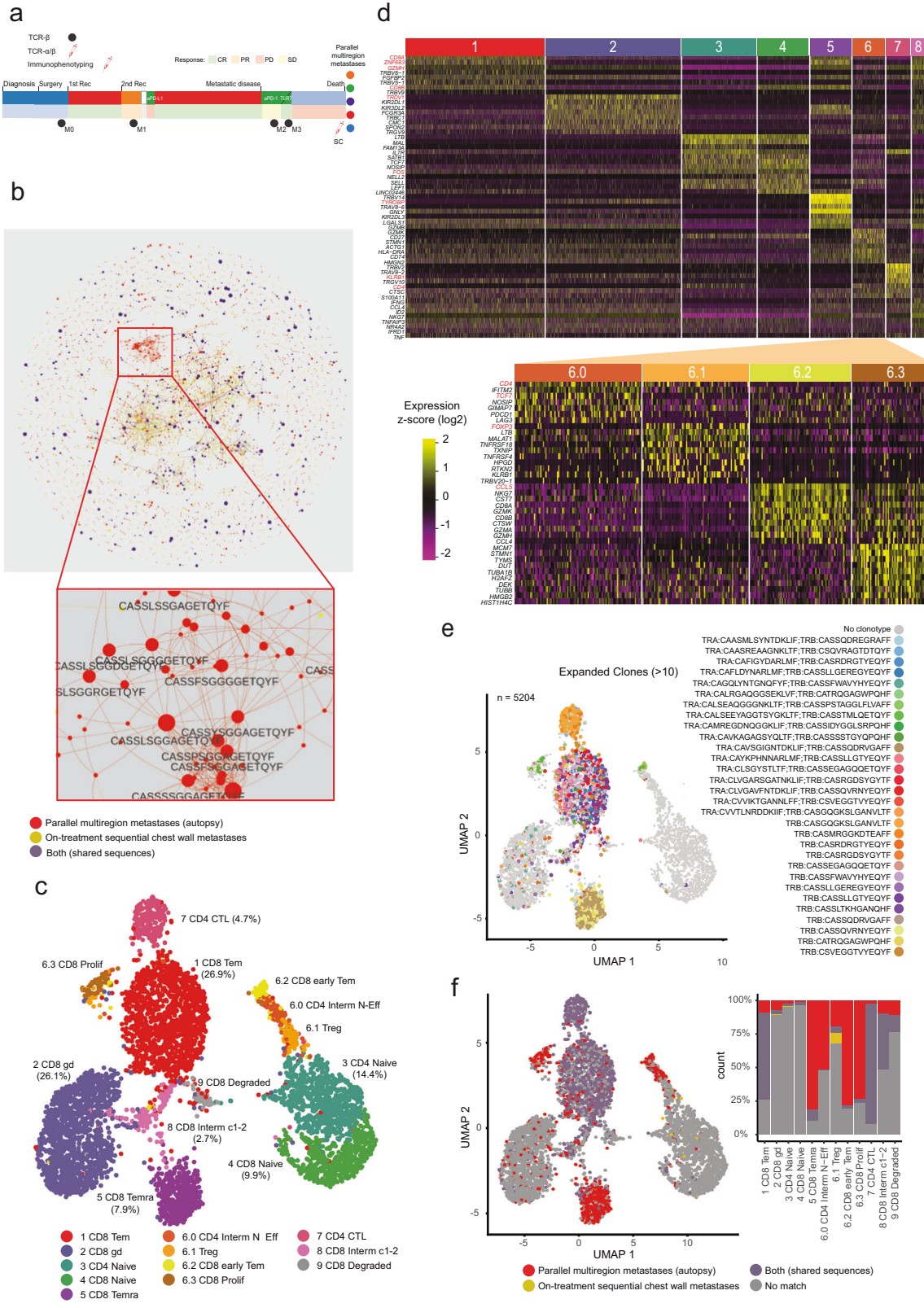

corroborate the IHC analyses from the TCRβ sequencing data, we inferred the fraction of T cells within the total nucleated cell count (range 0.007–0.225; *n* = 9 tumors, average = 0.071) as a surrogate measure of T cell infiltration. This analysis revealed that T cell fraction in tumor tissue gradually decreased over time (Mann-Kendall test, *p*-value = 0.0261) as metastatic disease evolved, suggesting that early metastases were more infiltrated by T cells than metastases at autopsy

(Supplementary Fig. 2i). In parallel, clinical peripheral blood samples of the TNBC index patient taken from day ~850–2033 showed leukocyte counts within normal rage, but progressive lymphopenia over time and high neutrophils-to-lymphocytes ratio just before death (day 2033) (Supplementary Table 2). These results suggest that a tumor-specific response was being mounted at the periphery during the entire course of the disease with some traffic of T cells from blood to

**Fig. 2 | Sequential TCR repertoires evolve over time leading to mixed dysfunctional single T cell states. a** Workflow of sequential TCRβ detected from nine metastases: 4 on-treatment (M0-day 373, M1-day 799, M2-day 1687, M3-day 1687) at distinct time points and 5 parallel multiregion postmortem metastases. scRNA-seq and scTCR-seq analyzed in isolated T cells from peripheral blood (day 2031). aPD-L1, anti-programmed death-ligand 1 monoclonal antibody; aPD-1 anti-programmed cell death protein 1 monoclonal antibody, CR complete response, M metastasis, PR partial response, PD progressive disease, Rec recurrence, SD stable disease, SC single-cell, TCRα T cell receptor alpha chain, TCRβ T cell receptor beta chain, TLR7 Toll-like receptor 7. **b** TCRβ CDR3 repertoire joined network to elucidate subnetworks private to on-treatment sequential chest wall metastases (yellow) or to parallel multiregion metastases (red) or shared to both sets of metastases (purple). Insert on the bottom shows amino acid sequences from a parallel multiregion metastases-specific subnetwork. Each node's size corresponds to the number of samples where the sequence has been detected. Edges were formed between nodes only when the edit distance between the two CDR3 sequences equaled 1. Source data are provided as a Source Data file. **c** Uniform manifold approximation and projection (UMAP) analysis that displays single-cell transcriptomic landscape of sorted CD3+CD19- single T cells. Single T cells are colored by expression cluster,

based on gene expression difference, of 11 T cell subsets and functional states. Mean unique molecular identifier (UMI) counts per cell were 3726 with a median number of genes detected per cell of 1254 (98.92% CD3+ cells of all cells, 97.17% expressing CD3ε, 89.83% expressing CD3δ). Clusters with percentages above 2% are depicted. Source data are provided as a Source Data file. **d** Heatmap from the scRNA-seq showing 9 clusters of T cell subpopulations resolved by z-scored differential expression of curated T cell marker genes. Caption shows four subclusters integrated within cluster 6. The top markers that define each one of those clusters are highlighted in red. Cluster 9 was characterized by cells with few detected genes and a high fraction of mitochondrial counts indicative of damaged cells. Source data are provided as a Source Data file. **e** UMAP embedding single cells from peripheral blood showing TCR clonotypes classified as singletons or expanded ($N = 5204$ cells with TCR) were projected onto the UMAP of peripheral blood T cells. Source data are provided as a Source Data file. **f** UMAP embedding single T cells from peripheral blood ($N = 1284$) and barplot colored by TCR clonotypes found in on-treatment sequential metastases (yellow), parallel multiregion metastases (red) or present in both tumor sources (purple). Source data are provided as a Source Data file.

tumor tissues, although the percentage of infiltrating T cells declined steadily as the patient progressed.

Taken together, our results document the phenotypic diversity of peripheral blood T cells close to death, including CD4/CD8 T cell phenotypes still with some cytotoxic potential and enrichment for γδ. Even close to death, the immune system appears to be still mounting an adaptive immune response against parallel multiregion metastases while keeping T cell memory clonotypes against early responding metastases at the periphery. The TCR repertoire, however, was evolving and losing complexity over time, with convergent evolution into similar immune receptor clonotypes. This could be due to a reduced neoantigen exposure over time due to different immune evasion mechanisms, resulting in a less diverse, more focused response at the end of the TNBC patient's life. These prompted us to explore the status of the immune microenvironment and the genomic-based neoantigen heterogeneity in tumor tissue to understand the likely mechanisms of immune evasion.

## Changes in IFNy signaling, antigen processing machinery signatures, and allelic-specific HLA imbalance parallel metastatic tumor progression

To determine whether there are differences in the tumor microenvironment (TME) along the disease course and within parallel metastases, we applied a 770 immuno-oncology and proliferation gene panel[14,15] to all tumor specimens of the TNBC index patient ($n = 32$). Two main clusters were revealed, one defined by upregulation of immune signatures (hot or inflamed cluster), which included most of the on-therapy early metastases (from day 373 to day 1687), and another one defined by the enrichment of proliferation and hypoxic signatures (cold cluster), including all the metastases at autopsy (day 2033; Fig. 3a). RNA-seq based CIBERSORT, which deconvolves 22 immune cells, further highlighted the degree of immune cell heterogeneity within metastases and primary tumors of the index case and the TNBC cohort (Fig. 3b).

A comparison of the gene expression signatures of temporally earlier chest wall tumors ($N = 3$, day 373) to late chest wall metastases ($N = 4$, day 2033) of the TNBC index patient revealed downregulation of immune signaling pathways, in the late chest wall tumor samples, including IFNy signaling, members of the MHC/HLA class I and II loci, immune checkpoints (CTLA4, false discovery rate) (FDR < 0.02), TIGIT (FDR < 0.003) and inflammatory chemokines (CXCL9, FDR < 0.0013; CXCL13, FDR < 0.0052) (Supplementary Fig. 3a). Downregulation of these inflammatory pathways in late metastases might account for the reduced presence of infiltrating T cells in these tumors, which was confirmed by an enrichment of cytotoxic T cells in the sequential early

samples and a decline in late metastases, according to pathology CD8 T cells ($p$-value = 0.042) and CD4 T cells ($p$-value = 0.059) analyses (Supplementary Fig. 3b).

Next, we explored the temporal evolution of IFNy signaling, HLA class I-related APM, and cytotoxic T lymphocyte (CTL) abundance along with tumor proliferation over clinical responses of the TNBC index patient. IFNy signaling and APM signatures displayed dynamic changes over time, with their sharp upregulation correlating with clinical responses and a substantial reduction throughout later systemic treatments (non-responsive metastases) and postmortem parallel multiregion chest wall metastases (Fig. 3c). Conversely, the tumor proliferation signature, which may be considered a surrogate measure of tumor growth, displayed the opposite dynamics (Fig. 3c, Supplementary Fig. 3c). CTL abundance followed a similar trend to IFNy, with increasing rates up to day 799, before showing a consistent decrease toward later time points (Fig. 3c). These results are consistent with the histological data, showing elevated CD8 and CD4 T cell markers staining in early metastases and the declining T cell fraction estimates retrieved from the TCRβ sequencing data (Supplementary Figs. 2i and 3b). We observed a positive correlation between pathology tumor-infiltrating lymphocytes (TILs) estimates and mRNA-based T cell estimates, validating the accuracy of the gene expression-based inference of T cell abundance (Supplementary Fig. 3d).

To understand the gradual temporal decline in anticancer immunity, we defined the immunophenotype status that is, inflamed (IFNy $^{high}$, APM$^{high}$), desert (IFNy$^{low}$, APM$^{high/low}$), excluded (IFNy$^{high}$, APM$^{low}$) of sequential tumor biopsies and 20 parallel multiregion metastases of the TNBC index patient employing the IFNy signaling and APM signatures. The immunophenotype score allowed us to assign an inflamed immunophenotype to the sequential on-treatment tumors, and a desert immunophenotype to the majority of parallel metastases (Fig. 3d). Interestingly, two chest wall metastases (M17 and M18) displayed higher APM signature scores and were mapped as inflamed, unlike the rest of parallel metastases. Conversely, a tumor biopsy taken on-treatment (M3.1) and considered refractory to therapy, was mapped as excluded. We then validated the immunophenotype status with a Multiplexed Proteomic Assay (IFNy and MIC A/B (aka HLA-A and B)) across 35 samples in which 19 were parallel multiregional metastases that mapped mostly to the desert phenotype (IFNy$^{low}$, MIC A/B$^{high/low}$) (Supplementary Fig. 3e). The 14 on-treatment sequential samples from peripheral blood and cell-free DNA from pleural and ascitic fluids were tested for soluble IFNy and MIC A/B, and were mapped into either inflamed (IFNy$^{high}$, APM$^{high}$) or excluded (IFNy$^{high}$, APM$^{low}$) phenotypes. Histological data validated the enrichment of IFNy in earlier metastases as compared to late parallel

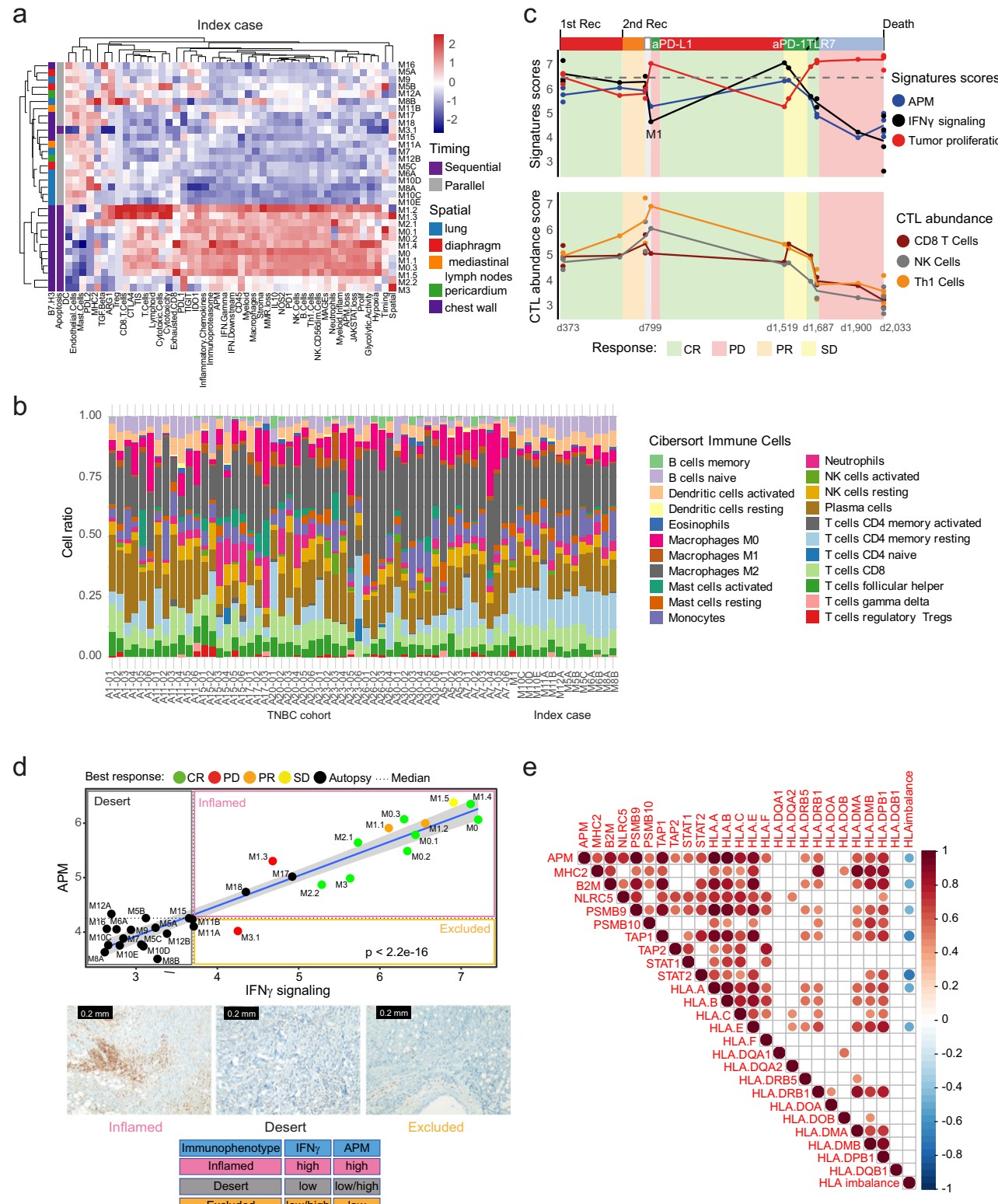

multiregion metastases (p-value: 9.971e-08, Wilcoxon rank-sum exact test) (Supplementary Fig. 3f).

Another possible mechanism of immune escape affecting the antigen presentation machinery might be mediated by an allelic imbalance at HLA loci, which has indeed been shown to reduce the capacity of neoantigen presentation, thereby aiding immune escape of tumors[16]. We employed an accurate HLA typing method for class I[17] and investigated the allele-specific HLA loss (HLALOH) and allelic

imbalance in sequential and parallel late metastases for HLA class I (Supplementary Fig. 3g). Of the TNBC index patient, four metastases had intact HLA haplotypes, M1 (day 799), a lesion that appeared earlier during the course of clinical follow-up, and M16, M17, M18, chest wall tumors that were sampled at autopsy, but categorized as inflamed using our immunophenotype score (Fig. 3d). Statistically significant haplotype loss (that is, HLA-A*11:01, HLA-B*27:05, HLA-C*02:02) was observed in non-responsive metastases categorized by desert

**Fig. 3 | Antigen-presenting machinery and immune signatures map metastases revealing immune escape. a** Unsupervised hierarchical clustering on gene signatures across sequential and parallel multiregion metastases. Gene expression for immune cell markers is represented by normalized log2 counts. **b** RNA-seq based CIBERSORT, which deconvolves 22 immune cells, was applied to both the index case (M1-day 799 chest wall metastases and 17 multiregion parallel metastases) and the TNBC cohort (10 primary tumors, 42 multiregion metastases). Source data are provided as a Source Data file. **c** Longitudinal monitoring of APM, IFNy signaling and tumor proliferation signature scores, cytotoxic T cell (CTLs) abundance, represented as per CD8 T cells, Th1 cells, and NK cells gene expression scores. Gene expression levels are represented by normalized log2 counts. An interferon-based tumor inflammation signature (TIS)[15] which integrates the IFNy signaling and APM signatures, was contextualized across 113 primary TNBCs of the TCGA dataset[73] (mean score 6.48). Gray dotted lines represent the median TIS score derived from these primary TNBCs of the TCGA. Source data are provided as a Source Data file. aPD-L1 anti-programmed death-ligand 1 monoclonal antibody, aPD-1 anti-programmed cell death protein 1 monoclonal antibody, CR complete response, d day, M metastasis, PR partial response, PD progressive disease, Rec recurrence, SD stable disease, TLR7 Toll-like receptor 7. **d** Immunophenotype status, defined by IFNy signaling and APM signatures, mapped as inflamed (IFNy high, APMhigh), desert (IFNylow, APMhigh/low), excluded (IFNyhigh, APMlow). The median value of APM and IFN signatures relative to the immunophenotype distribution is defined as the cutoff for the stratification. The blue line represents the linear regression and its 95% confidence interval for the APM signature in function of the INFy signaling expression. Source data are provided as a Source Data file. Representative micrographs of inflamed, desert, and excluded tumors are shown below the immunophenotype map to display a metastasis with a high level of immune infiltration (inflamed), immune cell accumulation but not efficiently infiltrated (excluded), low/absent level of immune infiltration (desert). Scans in 40× objective, scale bar 0.2 mm in each IHC panel. **e** Spearman correlation matrix of HLA-related gene expression profiles and number of imbalanced HLA class I alleles (0, 1, 2, and 3) among 20 parallel multiregion metastases of the index case. Red color represents positive correlation whereas blue represents negative correlations. Color intensity and size of the circle are proportional to the correlation coefficients, which are depicted in the legend to the right. Blank squares correspond to non-significant (p-values > 0.05) correlations. Source data are provided as a Source Data file.

immunophenotypes, whereas metastases with intact HLA or with a single allele imbalance displayed inflamed phenotypes. In the extended cohort of 8 metastatic TNBC patients, we determined the status of HLA class I in primary tumors and multiregion metastases (Supplementary Fig. 3g). Here, the primary sample of patients A7 and A20 had intact HLA haplotypes; by contrast, primary tumors A1, A5, A11, A15, A17 had imbalance and A11 presented HLALOH. Metastases were heterogeneously affected by imbalance, HLALOH, or HLA integrity.

We next correlated the HLA-related gene expression profiles and HLA allele-specific imbalance among the 20 parallel multiregion metastases of the TNBC index patient. We observed that the greater the HLA allelic imbalance, the lower the gene expression of HLA class I-related genes (HLA-A, HLA-E, B2M, TAP1), IFNy pathway genes (STAT2[18], signal transducer and activator of transcription), a major component of the immunoproteasome[19](PSMB9), and the APM signature (Fig. 3e).

Taken together, our results show that metastatic tumor progression is paralleled by a decrease in IFNy signaling, APM, and inflammatory chemokines, a reduction in T cell infiltration along with an impairment of specific allelic HLA integrity.

## T cell exhaustion score and soluble PD-L1/IFNy define the immune evolution during metastatic progression

T cells are exposed to persistent antigen and/or inflammatory signals posing a scenario that is often associated with the deterioration of T cell function leading to T cell exhaustion. We developed a T cell exhaustion metagene score comprising the most common immune checkpoint inhibitory molecules (PDCD1 (PD-1), LAG3, TIM3 (HAVCR2), KLRG1, TIGIT, CD244, CD160, BTLA, CTLA4, ENTPD1, CD160, ID2). The T cell exhaustion metagene score was based on the weighted arithmetic mean using the log fold-change of differentially expressed genes and was set up between multiregion metastases and primary TNBC breast tumors of the Siegel dataset[10] as coefficient. The T cell exhaustion metagene score was applied to the specimens of the TNBC index patient (M1-day 799 chest wall metastasis, obtained before treatment with the immune checkpoint inhibitor atezolizumab, and 17 parallel multiregion metastases) and the extended cohort of 10 TNBC patients[10] (10 primary tumors and 42 multiregion metastases) using RNA-seq based gene expression levels.

We quantified the T cell exhaustion metagene score for each sample individually (Fig. 4a, Supplementary Fig. 4) and observed a positive correlation between the T cell exhaustion score and the cytolytic score, defined as the geometric mean of the GZMA and PRF1 expression levels (rho = 0.52; p < 0.002), with the presence of cytotoxic T cells immune gene expression signature (rho = 0.35; p = 0.002, (Fig. 4a, Supplementary Fig. 4)).

Taken together, the TNBC index patient and the extended cohort of TNBC patients, most of metastatic lesions (79.4%) showed a negative score and 80% of primary tumors had positive scores (p = 0.0005), denoting an inflamed to desert immune evolution during metastatic progression (Fig. 4a, Supplementary Fig. 4). We observed that the majority of parallel multiregion metastases were assigned as "desert tumors" (T cell exhaustion metagene scoreLow/ cytolytic scoreLow / CD8+Low) further validating the gene expression-based immunophenotype (IFNlow, APMhigh) (Fig. 3d) and the Multiplexed Proteomic Assay (IFNlow, MIC A/Bhigh/low)(Supplementary Fig. 3e) "desert" immunophenotype.

Of clinical interest, we profiled soluble PD-L1 and IFNy in 14 sequential liquid biopsies in plasma or serum of the TNBC index patient and integrated results with the gene expression-based PD-L1 levels of the on-treatment sequential metastases (Fig. 4b). Of note while following the dynamics of soluble and tissue-based PD-L1 and IFNy at the time of major clinical responses, we observed that a sharp increase in soluble PD-L1 in plasma provided a better proxy at tumor progression to atezolizumab, indicating a mechanism of immune evasion that was only captured as biomarker in the liquid biopsy. Following clinical complete response, the levels of soluble PD-L1 declined, while IFNy levels increased, in line with a complete clinical response to the administration of cisplatin plus gemcitabine.

Therefore, the quantification of T cell exhaustion status of each metastasis based on immune checkpoint inhibitory molecules and the use of non-invasive soluble PD-L1 and IFNy levels guided the antitumor immune evolution of the TNBC index patient during metastatic progression.

## Increased genomic complexity in mutation and neoantigen may trigger immune escape

To explore if an increase in the genomic complexity of metastases contributes to a decline in antitumor immunity, we decomposed the mutational repertoire and copy number alterations of the primary breast cancer (P0-day 0), chest wall lesions (M0-day 373, M1-day 799, M3-day 1687) and 20 parallel multiregion metastases affecting different anatomical sites (day 2033) of the index TNBC patient. We inferred the clonal structure and genomic subclonal heterogeneity of each specimen with targeted sequencing and/or whole-exome sequencing (WES) and observed inter-lesion heterogeneity within and between primary breast cancer, early sequential chest wall metastases and parallel multiregion metastases (Fig. 5a, Supplementary Fig. 5a).

We quantified the tumor mutation burden (TMB) of metastases of the TNBC index patient, defined as the number of non-synonymous somatic mutations per megabase of interrogated genomic sequence, and predicted putative neoantigens corresponding to expressed

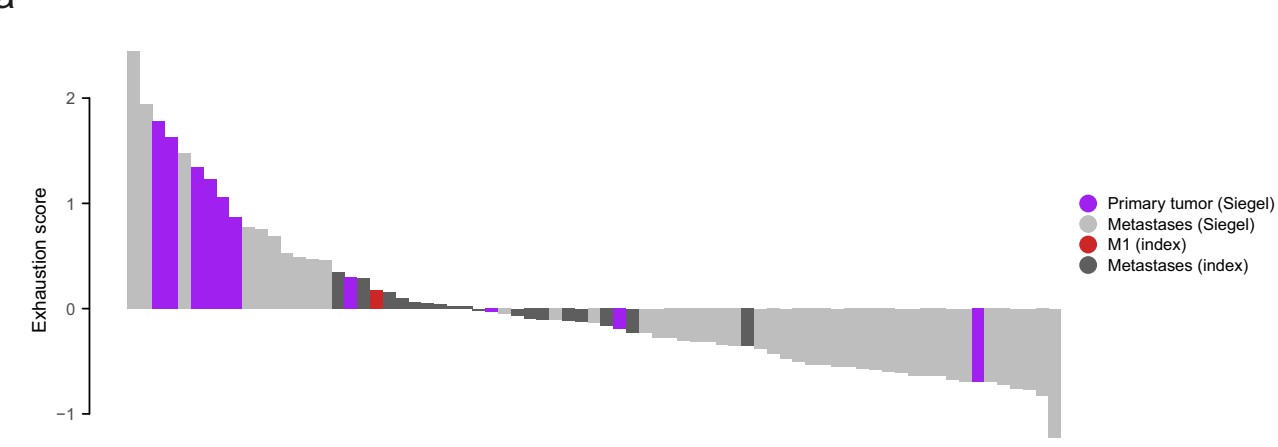

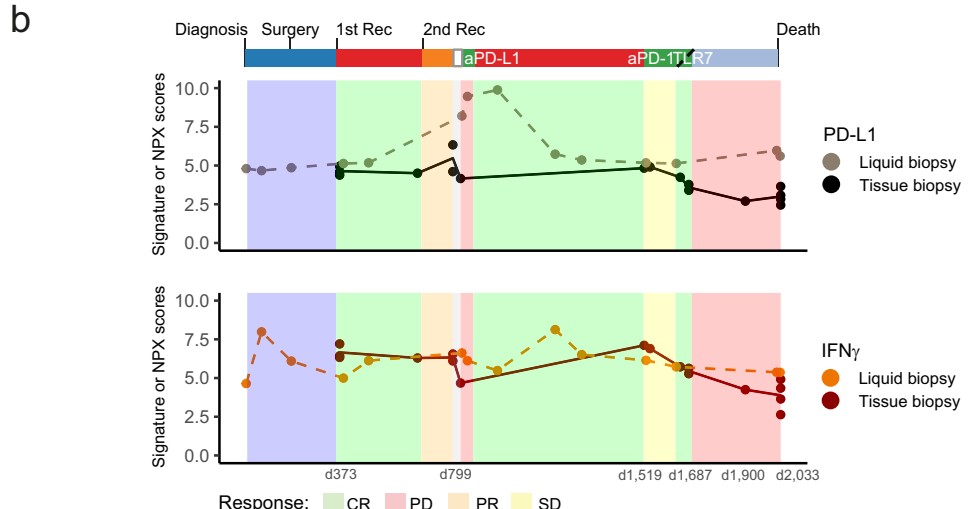

**Fig. 4 | T cell exhaustion score and soluble PD-L1 and IFNγ follow the immune evolution during metastatic progression. a** Barplot showing quantification of T cell exhaustion metagene. The T cell exhaustion metagene score was applied to the specimens of the TNBC index patient (M1-day 799 chest wall metastasis, obtained before treatment with an immune checkpoint inhibitor atezolizumab, and 17 parallel multiregion metastases) and the cohort of 10 TNBC patients (10 primary tumors and 42 multiregion metastases) that have RNA-seq gene expression available. Source data are provided as a Source Data file. **b** Longitudinal monitoring of soluble PD-L1 and IFNγ as liquid biopsies (proteomic level) and in tissue (gene expression scores). Clinical responses are depicted. Source data are provided as a Source Data file. aPD-L1 anti-programmed death-ligand 1 monoclonal antibody, aPD-1 anti-programmed cell death protein 1 monoclonal antibody, CR complete response, d day, PR partial response, PD progressive disease, Rec recurrence, SD stable disease, TLR7 Toll-like receptor 7.

mutations derived from cancer cells that may stimulate T cell immune responses. Parallel multiregion metastases showed increased TMB (median, 3.76 mutations/Mb) and neoantigen burden (NB, median, 69 neoantigens) as compared to earlier specimens (TMB: 2.52 and 3.51 mutations/Mb in M1-day 799 and M3-day 1687 chest wall metastases; NB: 3 in P0, 1 in M0, 3 in M1, and 23 in M3 (range in all samples 1–151)) and relative to the TCGA TNBC[8,20] primary cancers (1.7 mutations/Mb, 105 samples, *p*-value = 8.8e-8) suggesting acquisition of mutations and neoantigens during metastatic TNBC evolution (Fig. 5a, Supplementary Fig. 5b).

Phylogenetic trees depicted the sequential and spatial clonal evolution of the index TNBC patient and identified the driver mutations in each branch (Fig. 5b). A frameshift mutation in *TP53* (T256fs) was found to be a truncal driver mutation present in all metastases and plasma cell-free tumor DNA samples. This mutation was validated using deep targeted sequencing and digital PCR in selected tumors as well as in plasma cell-free tumor DNA at different time points, from

diagnosis until death (Supplementary Fig. 5c). Two neoepitopes generated from *TP53* T256fs mutation were computationally predicted (RRPILTIINT for HLA-B*27:05, IINTGRLQW for HLA-C*02:02) and tested for their ability to give rise to functional T cell-mediated immune responses in the tumor host. We designed a set of 21 peptides based on these two neoepitopes. Ex-vivo IFNγ ELISpot assay using autologous peripheral blood mononuclear cells (PBMCs) from day 2031 (48 h before death) revealed this clonal driver mutation to generate bona fide immunogenic neoantigen epitopes, with a specific 11mer (RRPIL-TIINTG) being most immunogenic peptide (Fig. 5b). Notably, our results demonstrate that the *TP53* T256fs-specific T cell response in peripheral blood was still present at the end of life of the patient when the breast cancer was highly proliferative.

To define how increased heterogeneity can trigger immune escape, we interrogated the time course of mutations (Mutation Time) relative to copy number gains using WES data and a molecular clock analysis[21] across the primary breast cancer (P0), the sequential (M0,

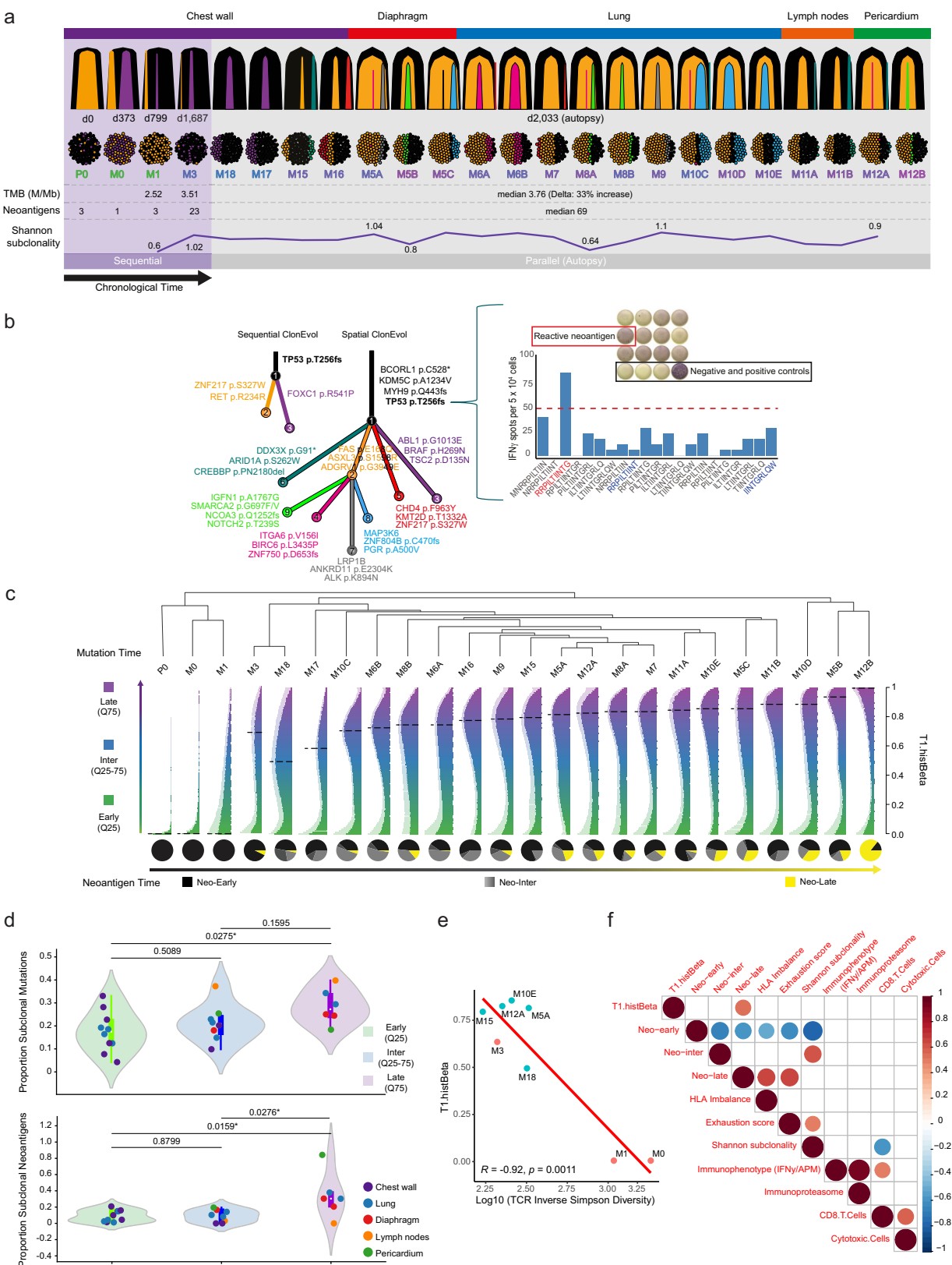

M1, and M3) and 20 parallel multiregion metastases of the TNBC index patient (Fig. 5c, "Methods" section). We considered somatic mutations relative to clonal and subclonal copy number states to classify mutations into 3 different time transitions: early, intermediate, and late (see Methods).

The Mutation Time (T1.hisBeta) defined each sample's molecular time and was used to molecularly order sequential and parallel multiregion metastases. As expected, the primary breast cancer-P0, and temporally earlier chest wall metastases (M0-day 373, M1-day 799, and M3-day 1687) had an early Mutation Time, reinforcing the robustness

**Fig. 5 | Clonal architecture and Neoantigen timing of sequential and parallel metastases of the index patient. a** Clonal architecture and timing of sequential and parallel metastases of the index patient. Bell plots showing the clonal composition and evolution of metastases in time and space. Clustering of mutations by cancer cell fraction (CCF) among metastases was performed by PyClone-vi, resulting in 9 clusters, where mutations in cluster 1 were inferred as clonal. The clonal admixture inferred from each metastasis is represented. Source data are provided as a Source Data file. Tumor mutation burden (TMB) (mutations per megabase, M/Mb), the number of neoantigens and the Shannon subclonality index are shown for chest wall tumors and parallel multiregion metastases. **b** TNBC patient clonal history is shown by sequential (temporal) and spatial phylogenetic trees, where the nodes represent the clones; branches represent evolution paths (length scaled by the square root of number of clonal marker mutations). Branches are labeled with potential driver mutations, and clone nodes are labeled with cluster identification. Source data are provided as a Source Data file. A frameshift mutation in *TP53* (T256fs) was characterized as a truncal driver mutation present in all metastases. T cell reactivity against *TP53* T256fs by IFNy ELISpot of PBMCs (day 2031) is shown. Thresholds for positive responses were determined as at least five spots (50 SFC/106 PBMCs) after background subtraction. **c** Molecular clock hierarchical clustering analysis depicting Mutation Time (y-axis, defined by T1.histBeta) for sequential P0 (primary), M0-day 373, M1-day 799, M3-day 1687 chest wall tumors, and 20 parallel multiregion metastases. Histograms illustrate the distribution of event timing within these samples, categorized as early (lowest quartile, Q25), intermediate (inter, Q25–75), or late (highest quartile, Q75) based on the criteria detailed in the methods section. Source data are provided as a Source Data file. Pie charts depict the distribution of the inferred Neoantigen Time: neo-early (%

neoantigens that appear early on), neo-inter, and neo-late (% neoantigens that appear late on). **d** Violin plots of subclonal mutations among early, inter, and late metastases, depicting the proportion of subclonal mutations (top) and subclonal neoantigens (bottom), respectively. $N = 24$ biologically independent samples in both the upper and lower panels, representing primary tumor, sequential, and parallel multiregion metastases. Statistical analysis among groups made by one-way ANOVA with Tukey's test for multiple comparisons is shown. Boxplot limits indicate the interquartile range (IQR; 25th–75th percentile), with a center line indicating the median. Whiskers show the value ranges up to $1.5 \times$ IQR above the 75th or below the 25th percentile with outliers beyond those ranges shown as individual points. The color of metastases refers to the organ of origin. Source data are provided as a Source Data file. **e** T1.hisBeta, used to define each sample's molecular time (i.e. Mutation and Neoantigen Time) and to order sequential and parallel multiregion metastases, was negatively correlated to TCR Inverse Simpson Diversity (Log 10) ($p$-value = 0.0011). Pearson correlation test, two-sided, no adjustments were made for multiple comparisons. $P$-value < 0.05 is considered statistically significant. Source data are provided as a Source Data file. **f** Integration of key immune and genomic parameters with Neoantigen Time. Spearman correlation test, two-sided, matrix of key immune (exhaustion score, gene expression immunophenotype, CD8 T cells, and cytotoxic cells) and tumor-related (HLA imbalance, Shannon subclonality) parameters and Neoantigen Time among sequential and parallel multiregion metastases of the index case ($N = 24$ biologically independent samples). Red color represents positive correlation whereas blue represents negative correlations. Color intensity and size of the circle are proportional to the correlation coefficients, which are depicted in the legend to the right. Blank squares correspond to non-significant ($p$-values > 0.05) correlations.

of our analyses (Fig. 5c). A Molecular Time hierarchical clustering analysis revealed two main branches (Fig. 5c). The first branch was defined by samples with early Mutation and Chronological Time (i.e. P0, M0-1). In contrast, the second branch represented a later origin, including metastases classified as late with high Mutation Time. The Mutation Time did not follow a linear, contiguous order per organ or multi-tissue site, potentially skipping contiguous areas of the lung, diaphragm, pericardium, mediastinal lymph nodes, or chest wall. There was an enrichment of subclonal mutations in metastases classified as "late" as compared to "early" ($p$-value = 0.02; Fig. 5d). These observations underscore the complexity of the evolutionary timeline and the genomic diversity inherent in this TNBC index patient. Similar chronological or molecular time origins exhibited varying genomic landscapes, resulting in increased heterogeneity across tumors and metastases.

We next explored the relationship of early and late neoantigens, present clonally or subclonally, with immune surveillance and immune escape. Therefore, we estimated the Neoantigen Time (T1.histBeta) and stratified samples of the TNBC index patient into three time-scale subgroups, referred to as neo-early, neo-inter, and neo-late (Fig. 5c, pie charts). Among postmortem parallel multiregion metastases, neoantigen heterogeneity varied, but showed a significant enrichment of subclonal neoantigens in metastatic lesions classified as neo-late ($p$-value = 0.0159; Fig. 5d). By contrast, early metastases in the chest wall (P0, M0, M1, M3), inflamed postmortem (M17 and M18, Fig. 3d) and specific lung cancer metastases (M6A, M6B, M8B, right lung; M10C, left lung) had the fewest subclonal neoantigens and, hence, were classified as neo-early (Fig. 5c).

Next, we integrated the three time-scale subgroups of Neoantigen Time with key immune-related factors. Firstly, we observed a strong negative correlation of T cell/TCR (Inverse Simpson diversity index) with neo-late ($r = -0.92$, $p$-value = 0.0011, Fig. 5e), indicating the adaptive T cell response, mounted to neoantigens exposure over time, not to be sufficient to counteract subclonal neoantigens. At the immune microenvironment, the immune exhaustion score negatively ($r = -0.60$; $p$-value = 0.006, Fig. 5f) and positively ($r = 0.67$; $p$-value = 0.002) (Supplementary Fig. 5d) correlated with neo-early and neo-late, respectively, suggesting late subclonal neoantigens to be present in the context of a less immune-proficient microenvironment.

Secondly, we analyzed the antigen-presenting machinery and observed a disrupted APM function, here represented by HLA imbalance, to be negatively correlated with neo-early, indicating that metastases with less capacity to present neoantigens appear later to further drive immune evasion. Of note, neo-late negatively correlated with APM ($r = -0.48$; $p$-value = 0.022) and IFNy ($r = -0.58$; $p$-value = 0.0044) from our immunophenotype (IFNy/APM) score, pointing to metastases with subclonal neoantigens to be less proficient in provoking an immune response (Supplementary Fig. 5e).

Thirdly, at the genomic level, we observed an increase in the proportion of subclonal neoantigens among metastatic lesions classified as of neo-late metastases. The Shannon subclonal diversity index negatively correlated with neo-early ($r = -0.46$, $p$-value = 0.034) and with CD8 T cells inferred from the Cibersort gene expression analyses ($r = -0.44$, $p$-value: 0.045). That indicates that a higher subclonal diversity later on could be associated with a less effective cytotoxic CD8 T cell responses.

Our analyses indicate that the dissemination of sequential to parallel metastases posed more genomic complexity via increased TMB and neoantigen subclonality. Specially, subclonal neoantigens that appear later during the evolution of the disease showed immune escape related to a less functional HLA machinery and an exhausted, immune-excluded microenvironment. Metastases with similar molecular time origins exhibited varying genomic and immune microenvironment landscapes. This intricate relationship underscores the need for precise molecular profiling and tailored treatments to address the diverse aspects of cancer evolution.

## Discussion

The major strength of our study lies in the continuous, deep monitoring of the antitumor immune responses to breast cancer metastases, highlighting the value of sequential and multiregion sampling to unravel the molecular timing and modes of immune escape.

We identified multiple immune escape processes, through the integration of genomic, transcriptomic, proteomic, molecular clock, and neoantigen analyses, complemented with immuno-histology and single-cell and bulk TCR repertoires. Distinct from previous studies showing independent mechanisms of immune escape at the genetic, epigenetic, or gene expression levels across patients in single

specimens or spatial-temporal analyses[8,22], our results demonstrate the importance of sequential multiregion samples to unravel that the immune responses against breast cancer are dynamic processes and reveal multiple evolving and convergent genomic and immune escaping mechanisms, *even within a single individual*.

Our results identify endpoint-reactive peripheral T cells with shared TCR repertoire of earlier and, in a larger scale, late multiregion tissue samples. The TCR repertoire, however, was evolving, showing loss of diversity and convergent evolution as breast cancer progresses. In this context, although T cells are expected to reach a dysfunctional or exhausted state in tumor tissues[23], we observed phenotypic diversity of also cytotoxic states, inferred from single-cell analyses of peripheral blood T cells sampled close to death. IHC analyses corroborated the presence of CD8 and CD4 cytotoxic T cells through the disease course but showed a steady decline over time until the patient expired. Even close to death, the immune system appears to be still mounting an adaptive immune response against synchronous metastases while keeping T cell memory clonotypes against early responding metastases at the periphery.

In line, a functional T cell response against the founder *TP53* T256fs-derived neoantigen was confirmed, although the patient evolved to a lethal phase, suggesting the specific response to be unable to halt tumor growth and progression. Because the presence of the HLA-B*27:05 allele is required for presentation of the *TP53* T256fs neoantigen, its allele-specific imbalance across selected parallel metastases might have enabled heterogeneous immune escape. Clonal neoantigens represent key targets for neoantigen-based vaccines and adoptive T cell therapies[24,25]. However, our findings suggest that the efficacy of targeting clonal neoantigens for universal clinical responses will depend on the degree of alternative immune escape mechanisms in a given patient.

We also demonstrated that the gene expression signatures in the IFNy signaling pathway and the APM mark progressing metastases over time into inflamed, desert, and excluded phenotypes. Our multiassay orthogonal analysis at the bulk and single-cell gene expression, genomic, and proteomic levels assisted in identifying the progression of on-treatment sequential to parallel multiregion metastases with exhausted/desert immunophenotypes and inefficient infiltration of productive T cells. We demonstrated that co-occurrence of allele-specific HLA class I imbalance, which was subjected to the selective pressures of the immune microenvironment, along with down-regulation of HLA processing machinery-related genes may have further contributed to a heterogeneous immune escape in sequential and parallel metastases.

Building on these observations, we developed a T cell exhaustion metagene score composed of the most common immune checkpoint inhibitory molecules. We then assigned primary, sequential, and multiregion metastases of the index patient and the extended TNBC cohort to inflamed or excluded immune status during metastatic progression. Furthermore, our multiplexed proteomic analyses targeting PD-L1 and IFNy as key soluble immune proteins in plasma, as compared to PD-L1 and IFNy gene expression signatures in tumor tissue, contributed to the longitudinal monitoring in the index patient and to better identify a potential pseudo progression to the immune checkpoint inhibitor atezolizumab, before complete clinical response to cisplatin-based chemotherapy was observed.

The dissemination of sequential to parallel metastases posed more genomic complexity via increased tumor mutational burden and enriched subclonal neoantigens in late metastasis. We developed Neoantigen Time to define the time course of neoantigens from the diagnosis to the end of life, to understand their contribution to heterogeneity and immune escape. In our study, the evolution and enrichment of subclonal neoantigen architectures revealed distinct routes, molecular timings, and molecular signatures of multiregion metastatic seeding, suggesting that progressive intra- and inter-metastasis heterogeneity to be a driving force behind an attenuated antitumor immune responses. Specifically, neoantigens that appear later in the course of the disease correlated with a less functional HLA machinery and deficient T cell activity and colder/exhausted immune tumor microenvironments.

Our work has limitations, which mostly stem from the small sample size and therefore lack of power to vertically detect associations across patients. Nonetheless, analyzing a considerable number of sequential and parallel multiregion specimens, allowed horizontal integration to pinpoint simultaneous immune escape mechanisms in a single-index patient with well-documented clinical responses to systemic therapy. To address sample size limitations, we used an extended validation cohort of 11 TNBC patients with primary and multiregion metastases, to confirm finding of the data-driven discovery phase. Another limitation consists in the fact we have not used integrated epigenomic data or microenvironment analyses beyond the immune TME to define alternative routes toward immune evasion, as previously reported[22].

Despite these limitations, we demonstrate that the interplay between breast cancer metastases and host antitumor immunity are concerted by distinct mechanisms of immune escape, even within the same patient. Using spatio-temporal multiomics analysis, we report that the tumor immune microenvironment shapes the evolution of TNBC under selective pressures and dictates clinical responses by promoting distinct and mechanistically convergent immune escape processes. Strategies to enhance the efficacy of immunotherapy will need to consider all the mechanisms of immune evasion in a single patient.

## Methods
### Sample collection and processing
Written informed consent to publish clinical information was obtained from the index patient and family under the research ethics committee of the Dexeus Institute of Oncology, Quironsalud Group, Barcelona. Research autopsy was performed under VHIO Warm Autopsy Program protocols approved by the institutional review board (IRB) of Vall d'Hebron University Hospital (Barcelona, Spain). All tissues, blood samples, and images in this study were obtained with the approval of institutional review board and patient´s and/or family´s consents.

A total of 112 specimens for the 12 patients were analyzed: 11 primary tumors, 15 on-treatment metastases, 18 serial blood samples, and 66 postmortem metastases. We evaluated the matched normal, primary tumor, sequential (longitudinal) archival, and frozen biopsies and blood samples during 2033-day follow-up of the index TNBC patient.

A rapid research autopsy was performed within 6 h of death by F.T. and allowed the sampling of twenty synchronous parallel metastases across five organs (lung, mediastinal lymph nodes, diaphragm, pericardium, and chest wall skin). This dataset included a total of 56 specimens, including a primary tumor specimen from diagnosis, 15 on-treatment sequential chest wall skin biopsies, 18 serial blood samples (plasma, serum, or whole blood samples (germline DNA and PBMCs)), 2 body fluids (ascitic and pleural) and 20 parallel multiregion metastases.

Specimens, including tumor tissue, were stored immediately in cryotubes in the autopsy room and then cryopreserved at −80 Celsius degrees for further experiments. Hematoxylin and eosin-stained tumor sections with cellularity percentage equal or higher than 10% were used in downstream experiments.

### RNA extraction for nCounter gene expression code set and RNA-sequencing (RNA-seq) of the index patient
Total RNA was isolated from 10-μm-thick formalin-fixed paraffin-embedded (FFPE) sections of 34 tumor samples with High Pure FFPE RNA Isolation Kit (Roche), according to the recommendations of the

manufacturer and quantified using fluorimeter with Qubit™ RNA XR Assay Kit (Invitrogen). RNA of the primary tumor was extracted directly from the archived slide-mounted FFPE tissue; this sample, however, did not pass the quality control step after nCounter gene expression analysis. RNA was extracted from 21 fresh-frozen tumors using RNeasy Tissue & Blood kits (Qiagen) for RNA-seq.

### DNA extraction of the index patient

DNA was isolated from 21 tumor and blood (germline) using DNeasy Tissue & Blood kits (Qiagen). Circulating cell-free DNA was obtained from 16 plasma samples using QIAamp ccfDNA/RNA Kit (Qiagen), according to manufacturer's specifications, and of these 9 passed quality control, with minimum 10 mg of DNA.

### Peripheral Blood Mononuclear Cell (PBMCs) isolation of the index patient

PBMCs were isolated from peripheral blood obtained 48 h before the autopsy procedure using Ficoll-Paque density gradient (GE Healthcare Bio) protocol prior to cryopreservation.

### Fluorescence-activated cell sorting (FACS) of the index patient

PBMCs were stained using CD3-APC-Cy7 (Biolegend), CD19- BV421 (Biolegend) antibodies and Live/Dead fixable violet dead cell stain kit (Thermofisher). CD3+ T cells were sorted using a BD FACSAria II (BD Biosciences) (Supplementary Fig. 5g). 10,000 live CD19- CD3+ T cells were used for Single T cell RNA and TCR Sequencing.

### TCR variable beta chain sequencing

Sequencing of the CDR3 regions of human TCRβ chains was performed using the immunoSEQ® Assay (Adaptive Biotechnologies, Seattle, WA) in 4 sequential on-treatment metastases and 5 parallel metastases of the index patient.

Repertoire analysis and diversity estimations were performed using immunArch with the provided by immunoSEQ ANALYZER (v2)[26]. To do a fair comparison and not biased the results by the varying number of clonotypes on each sample, these analyses were done after downsampling the number of clonotypes of all samples to 1000. Repertoire sharing strongly depends on the size of the repertoire. Downsampling has the goal to make data samples comparable and avoid size effects[26,27].

### Single T cell RNA-seq and T cell receptor (TCR) sequencing of the index patient

Cell concentration and viability of the single-cell suspension were verified using a TC20™ Automated Cell Counter. Cells were partitioned into Gel Bead-In-Emulsions (GEMs) by using the Chromium Controller system (10X Genomics) aiming at a Target Cell Recovery of 10,000 cells. Single-cell Gene Expression (GEX) and T cell receptor (TCR)-enriched libraries were prepared using the Chromium Single-Cell 5′ v1 Library and Gel Bead Kit (10X Genomics, Cat. N. 1000006) following the manufacturer's instructions. Briefly after GEM-RT clean up, cDNA was amplified using 14 cycles.

cDNA quality control and quantification were performed on an Agilent Bioanalyzer High Sensitivity chip (Agilent Technologies). Part of the quantified cDNA was used for the Human T Cell enrichment PCRs using the Chromium Single-Cell 5′ v1 Library Construction Kit (10X Genomics, Cat. N. 1000020) and Chromium Single-Cell V(D)J Enrichment Kit, Human T Cell (10X Genomics, Cat. N. 1000005) while the other part was used in the GEX library preparation. Both libraries were indexed by PCR using the PN-220103 Chromiumi7 Sample Index Plate. Size distribution and concentration of 5′ GEX libraries and TCR-enriched libraries, were verified on an Agilent Bioanalyzer High Sensitivity chip (Agilent Technologies). Finally, sequencing was carried out on an Illumina NovaSeq 6000 sequencer to obtain approximately 40,000 reads/cell, in the case of GEX libraries, and 2000 reads/cell for the TCR-enriched libraries.

### Single T cell RNA-seq and TCR data analysis of the index patient

The transcriptomic profiles and TCR genotypes were analyzed from each cell, in order to identify clonal immune cell populations within complex cell mixtures. Sequencing reads were mapped to hg38 reference genome and quantified through CellRanger (v.3.1.0). All downstream analyses were performed through Seurat suite (v.3.2.1)[28] for R v4.0.0. A quality control on the cells was applied prior to downstream analysis. Cells with <200 expressed genes, <1000 unique molecular identifier (UMI) counts, or >10% of expressed mitochondrial genes were excluded (likely degraded or broken cells). In addition, cells with >5000 expressed genes or >40,000 UMI counts were removed from the analysis to exclude potential doublets. Gene expression counts were log-normalized and scaled by regressing out gene counts and mitochondrial content. Degraded cells, which clustered separately, were filtered out by analyzing mitochondrial gene content (Supplementary Fig. 2a).

Cell clusters were identified in an unsupervised way using a community identification algorithm implemented in the "FindClusters" function (Seurat). Based on clustree optimization approaches[29] and using signatures from literature, we identified 11 clusters. Default setting was used with a resolution of 0.5. Cluster was annotated by determining differentially expressed genes (DEG) based on Wilcoxon rank-sum test. Cluster 6 displayed a mixture of markers from different cellular phenotypes. Therefore, we repeated the normalization, clustering, and DEG process specifically for this cluster. With this approach, we identified four subclusters that could be readily annotated with previous described markers and that were integrated with the initial clustering. TCR libraries were mapped to an enriched human TCR reference with CellRanger (v.3.1.0). TCR clonotypes were obtained from high-quality cells identified by CellRanger, containing full-length recombinant sequences and productive CDR3 chains.

### Degree of expansion

The Inverse Simpson Index was used to compare the degree of expansion across the samples. This value has been shown to be less sensitive to differences in sample size than other measures.

### Repertoire overlap

To compare the whole set of TCR regions, we measure the overlapping of public clonotypes by Morisita Index[13]. This index also takes into account the information about the abundances in the number of clones, giving a more realistic overlap than other measures like Jaccard Index[30].

### TCR Networks

To understand the evolution of T cell dynamics over time and between parallel multiregion metastases, we performed a network analysis using the TCRβ profiling of 4 on-treatment sequential chest wall metastases (M0-day 373, M1-day 799, M2-day 1687, M3-day 1687) and 5 multiregion metastases sampled at autopsy.

The networks were generated using the Levenshtein distance, a string metric for measuring the difference between two sequences. The distance of 1 amino acid change was set up as the threshold of similarity used to establish edges or connections in the network. Each node of the network is a unique TCR sequence. An edge between two nodes was created if the Levenshtein distance between the two sequences is 1.

To quantify the overlap between the on-treatment sequential chest wall metastases and the parallel multiregion metastases, we used three different metrics: network density, average clustering coefficient, and S metric. Network density describes the fraction of the

potential connections in a network that are actual connections between two nodes and shows how diverse a metastasis is (fewer connections implies fewer similar sequences in a given sample). The clustering coefficient, when applied to a single node, is a measure of how complete the neighborhood of a node is. We applied these analyses to the entire network and calculated the average of the coefficients of all nodes. This metric quantifies how connected the clusters or subnetworks are. Lastly, the S metric is defined as the sum of the products of degree(u) * degree(v) for every edge (u,v) in the network. The S metric measures the similarity of the network across each edge and connectivity of the nodes.

Next, we performed a joined network analysis, including on-treatment sequential chest wall metastases and parallel multiregion metastases, to detect subnetworks unique/private to either on-treatment metastases (yellow) or parallel multiregion metastases (red), or shared subnetworks (purple).

Lastly, to validate the robustness and stability of the metrics, we bootstrapped the samples by downsampling them to the number of cells of the smallest time points (M1-2915 cells) and reproducing the same analysis. This was done to reassure that the metrics do not depend on sample size since the variance in the number of cells between the samples is relatively big. The bootstrap was performed 10 times and showed that the metrics do not change significantly and do not depend on sample size.

### Immune cell type deconvolution

quanTIseq was used to quantify, via deconvolution of bulk RNA-seq data, the proportions of ten different immune cell types and the fraction of other uncharacterized cells present in the heterogeneous sample[31].

### PAM50 inference

The 50-gene subtype predictor PAM50 was applied to 11 primary TNBCs, 4 on-treatment metastases, and 66 postmortem metastases[10,11].

### Cytolytic activity score

Cytolytic activity score was calculated as the geometric mean of the GZMA and PRF1 expression levels from RNA gene expression data[32].

### Exhaustion score

A T cell exhaustion metagene score was developed based on a T cell exhaustion molecular signature comprising the most common checkpoint inhibitory molecules (PDCD1, LAG3, TIM3 (HAVCR2), KLRG1, TIGIT, CD244, CD160, BTLA CTLA4, ENTPD1, CD160, ID2). The T cell exhaustion metagene score is based on the weighted arithmetic mean using the log fold-change of differentially expressed genes between multiregion metastases versus primary tumors of Siegel dataset[10] as coefficient.

### Whole-exome and RNA-sequencing

DNA whole-exome libraries from 23 tumors and one germline sample were prepared with Agilent Human All Exon V6+COSMIC kit (Agilent) following manufacturer's instructions. RNA library preparation was performed with TruSeq Stranded Total RNA Library Prep Gold kit (Illumina) following the manufacturer's instructions. DNA and RNA libraries quality control were assessed with Bioanalyzer2100 (Agilent) and further quantified by qPCR, normalized, and multiplexed into a balanced pool. DNA-derived libraries were sequenced on an Illumina HiSeqX platform (2×150 paired-end chemistry) and RNA-derived libraries were sequenced on an Illumina NovaSeq600 platform (2×150 paired-end chemistry). Sequencing output of whole-exome sequencing (WES) and RNA-seq per library yielded 15 Gb (300×) and 200 M reads, respectively. Further, 3 sequential on-treatment tumor samples were subjected to WES at MSK's Integrated Genomics Operation[33], and

analysis were also conducted using our validated bioinformatics pipeline with the same parameters.

### Data preprocessing, alignment, and mutation calling

For the samples sequenced and processed in IrsiCaixa-Spain: Burrow-Wheeler Aligner (BWA) (v0.7.7-r441)[34] was used to align sequences with the latest genome assembly (GRCh38.p13)[34]. We called single nucleotide variant (SNVs) and INDELs using the Best Practices Work-flows of GATK (v4.1.2.0)[35] using MuTect2 (v4.1.0.0)[36] and added two extra filters in the function FilterMutectCalls (a) min-allele-fraction settled to 0.05 and (b) unique-alt-read-count to keep mutations supported by 10 or more reads. Somatic mutations were annotated using Variant Effect Predictor (v96.3)[37] and visualized using the Integrative Genomics Viewer (IGV) (v2.3.52)[38].

For the samples sequenced and processed in MSKCC-US: Reads were aligned to the reference human genome GRCh37 using the BWA (v0.7.15)[34]. The Genome Analysis Toolkit (GATK. V3.1.1)[39] was employed for local realignment, duplicate removal, and base quality recalibration. Somatic single nucleotide variants (SNVs) were detected with MuTect (v1.0)[36], indels with Strelka (v2.0.15)[40], Vars-can2 (v2.3.7)[41], Scalpel (v0.5.3)[42], and Lancet (v1.0.0)[43]. SNVs and indels outside of the target regions were filtered out, as were SNVs and indels for which the variant allele fraction (VAF) in the tumor sample was <5 times that of the paired normal VAF, and SNVs and indels found at >5% global minor allele frequency of dbSNP (build 137)[44]. Only somatic mutations with a depth ≥20 reads in the respective normal samples were considered[44]. All mutations were manually inspected using the IGV[38]. The cancer cell fraction (CCF) of each mutation was inferred using ABSOLUTE (v1.0.6)[45], as previously reported[46,47]. Mutations were cataloged as clonal if their probability of being clonal was >50%[48], or if the lower bound of the 95% CI of its CCF was >90%[49]. Copy number alterations and loss of heterozygosity were determined using FACETS[50]. LiftOverVCF from GATK[39] [Q] was then applied to anneal all variants in the same reference (GRCh38.p13).

### Targeted massively parallel sequencing analysis

7 out of 16 plasma DNA samples included in this study that passed quality control or had enough DNA for sequencing and 2 tumor tissues (Primary tissue and M0.1, sampled at D373) were subjected to targeted sequencing using the FDA-authorized Memorial Sloan Kettering-Integrated Mutation Profiling of Actionable Cancer Targets (MSK-IMPACT) assay[51], which comprises all coding regions and selected intronic and regulatory regions of 505 key cancer genes. Non-synonymous somatic mutations, amplifications, and homozygous deletions were retrieved from the original study. The raw MSK-IMPACT sequencing data (i.e. FASTQ files) were reprocessed using our vali-dated bioinformatics pipeline[47,52], for the inference of copy number gains and losses, loss of heterozygosity of genes targeted by somatic mutations, and mutational signatures. Mutations affecting hotspot codons were annotated. Non-synonymous TMB was calculated as the number of non-synonymous mutations divided by the total genomic region assessed by MSK-IMPACT, per megabase.

### Tumor mutation burden

TMB was defined as the number of non-synonymous somatic somatic per megabase of interrogated genomic sequence (for Agilent 65 Mb and for Nextera, 37 Mb). We analyzed the TMB estimations of TCGA cohorts using the same approach as reported in previous papers[53].

### TMB and gene expression purity-adjusted

To analyze the impact of tumor purity obtained from Sequenza, the TMB, gene expression profiles, and tumor purities were assessed using linear regression.

## RNA-sequencing data processing

FASTQ files of RNA-seq reads (paired-end) from tumor samples were pre-processed with Trimmomatic (v0.27)[54] to remove Illumina adapter sequences, trim low-quality read ends, crop long-reads to a maximum length, and discard short reads. Quantification of gene expression was performed with Kallisto (v0.46.0)[55] as transcripts per millions (TPM)[56] and raw counts.

## HLA typing

The 4-digit HLA class I and class II was called with the HLA-HD (v1.3.0)[17] tools using the WES-sequencing reads from the normal sample. To certify concordance, we compared HLA-HD calls of the normal sample versus all tumor samples of the TNBC case and determined that there was at least no two-digit HLA typing discrepancy between any normal-tumor pair for the index case.

## Neoantigen prediction

Whole-exome sequencing mutation calling, RNA-seq gene expression, and HLA typing were integrated as part of the neoantigen prediction pipeline. Tumor mutations were used to generate a comprehensive list of peptides (9–11 amino acids in length) with the mutated amino acid represented at each peptide position and used as input for machine-learning-based MHC-peptide binding predictors. For each non-synonymous coding mutation from a tumor, we predicted its impact specifically on the patient's HLA class I binding using the standalone version of the programs NetMHCpan-4.0[57] and NeoPredPipe(v.1.1)[58]. All 9–11-mer peptides containing the mutated amino acids were tested for binding to the patient's HLA-A, HLA-B, and HLA-C. A peptide was defined as a neoepitope based on two criteria, namely predicted binding affinity ≤500 nM and rank percentage ≤2% (default cutoff). Expressed non-synonymous mutations and neoepitopes were defined based on corresponding genes with normalized expression levels ≥5 TPM.

## IFNy ELISpot assay using ex-vivo PBMCs of the index patient

IFNy ELISpot assay ($8.5 \times 10^4$ PBMC/well) was performed to assess T cell antitumor immunity to neoantigens, using T cells expanded with IL-2 supplementation (50 IU/mL) for 15 days from cryopreserved isolated PBMC[59]. The stimuli was a set of 21 peptides (9, 10, and 11 mers) covering all positions of the *TP53* T256fs mutation[60,61]. The magnitude of the response (spot forming cells (SFC) per $10^6$ PBMC) was recorded.

## Shannon–Wiener diversity index

We defined Shannon–Wiener diversity index to characterize the sub-clonal diversity of metastases as

$$H = -\sum_{i=1}^{s}(p_i \log_2 p_i) \tag{1}$$

where **s** is the number of clonal clusters and $\mathbf{p_i}$ is the proportion of the community represented by that cluster.

## Analysis of Mutation and Neoantigen Time

For the Mutation and Neoantigen Time analyses, and for each tumor sample, somatic events can be timed relative to one another with different certainty. Subclonal events occur in a subpopulation of cells, and thus occur at a later point in tumor development than clonal events, which occur in all cancer cells in the sample population. The likelihood that an event is clonal or subclonal was considered. We timed somatic mutations relative to clonal and subclonal copy number states and calculated the relative timing of copy number gains using the R package MutationTimeR (v1.00.2)[21], ran with default parameters.

This method calculates the Mutation Time for each somatic mutation based on the relative copy number state for the location of the variant. Time is measured as a fraction of point mutations; this is termed *Mutation Time*. Mutation Time is proportional to real-time if the number of mutations acquired per bp and per year is constant[21].

We adapted the MutationTimeR algorithm, used in WES data, performing timing of mutations relative to gains to classify mutations in 4 different timing stages as clonal early, clonal not specified, clonal late, and subclonal.

These 4 states produce 3 different time transitions we analyzed: (i) early (ie., clonal early, lowest quartile of the time-scale) referred if the mutation occurred preceding or after the copy number gains, (ii) intermediate (ie. clonal late, clonal not specified, intermediate quartiles) referred if the mutation occurred after the copy number gains or if the mutations are present in all tumor cells, and (iii) late (ie., subclonal, top quartile) referred if the mutations are present only in a fraction of the tumor cells.

Therefore, we calculated Neoantigen Time akin to Mutation Time and fitted each metastasis into one of three categories, defined based on whether they represent early as defined by the lowest quartile of the time-scale, inter (intermediate quartiles), late (top quartile) events in breast cancer evolution.

The value of T1.hisBeta was calculated from the mode of the first copy number gains time and defined each sample's molecular time (aka Mutation and Neoantigen Time) and ordered parallel multiregion metastases. The dendrogram was constructed using the 'clustermap' function from the Seaborn[62] library in Python, which executes hierarchical clustering. In this analysis, timed copy number variants derived from MutationTimeR served as the input dataset and default function parameters.

## Analysis of clonality and reconstruction of clonal evolution

We used Sequenza(v3.0.0)[63], PyClone-VI(v0.1.0)[64], and ClonEvol bioinformatic[65] tools to determine the clonal ordering and clonal evolution models for the synchronous metastases. For each case, variant calls were integrated with local allele-specific copy number (obtained from Sequenza), tumor purity (also obtained from Sequenza), and variant allele frequency. All mutations were then clustered using the PyClone-VI allowing for up to 10 clusters (clones) out of 15 tested to best fit the evolutionary model of ClonEvol, using a binomial distribution, performing 1000 random restarts and 100,000 iterations and remaining parameters set as default. Clusters were inferred based on their mutation variant allelic fraction in order to identify founding clones and subclones across all metastases. Subsequently, we applied ClonEvol[65] to infer clonal evolution models, ignoring one cluster with no variability across samples, and set to default parameters.

## Droplet digital PCR (ddPCR)

The QX200 Droplet Digital PCR (ddPCR™) System (Bio-Rad Laboratories, Hercules, CA) was applied to chest wall biopsies and liquid biopsies of the index case to detect the clonal *TP53* T256* frameshift variant and corresponding wild-type alleles. Two nanograms of DNA per sample were used for digital PCR analysis. The 20 µL final volume of TaqMan PCR reaction mixture was assembled with 1× ddPCR Supermix for Probes (no dUTP), custom primers/probes assay for target (FAM) and wild-type (HEX) (900 nM of each primer, 250 nM of each probe) and 2 ng of genomic DNA templates (8 µL). Each assay (*TP53* T256* frameshift) was performed in separate mixes and loaded 96 well plate. The generated droplets were thermal cycled with the following conditions: 5 min at 95 °C, 40 cycles of 94 °C for 30 s, 52 °C for 1 min followed by 98 °C for 10 min (Ramp Rate 2 °C/s). After PCR, droplets were read in the Droplet Reader for fluorescent measurement of FAM and HEX probes and analyzed with QuantaSoft version 1.7.4. Mutation Allele Frequency was calculated as follows: MAF = (Nmut/(Nmut + Nwt)), where Nmut is the number of mutant events and Nwt is the number of WT events per reaction. The ddPCR analysis of solid

tumor DNA template and no DNA template were included as positive and negative controls, respectively. Only samples with >10,000 droplets were accepted as valid. Samples in clinical time points of low tumor burden were done in duplicate.

### Sequential-parallel tumor gene expression profiling of the index patient

50 ng of total RNA isolated from sequential biopsies (n = 13) (12 FFPE sections and 1 fresh-frozen) and metastases (n = 19) taken at autopsy were successfully hybridized to the code sets of NanoString®PanCancer Immuno-Oncology IO 360 Panel (770 genes), according to the recommendations of the manufacturer. The hybridized samples were run on the NanoString nCounter Preparation Station and scanned on the NanoString nCounter Digital Analyzer. The nSolver analysis package was used for quality control measures, log2-transformed data normalization, and gene expression profiling (GEP) analysis. Immune-oncology signature scores were obtained with nCounter using algorithms that summarizes a combination of expressed genes, which were computed after normalization to 10 housekeeping genes. The NanoString® PanCancer BC v2 360 Panel (776 genes) was applied and used to infer TNBC subtypes.

### HLA class I LOH

To determine whether maintenance or loss of HLA were present in multiregion metastases, we used LOHHLA[11,16] applying default settings to determine allele-specific copy number of HLA locus. At each heterozygous HLA locus in germline, LOH was inferred if the copy number for one of the two alleles was below 0.5 and log copy ratio difference between the two alleles was statistically significant (PVal_unique <0.05). Allelic imbalance is determined if $p < 0.01$ using the paired Student's $t$-test between the pairwise difference in logR values at mismatch sites between the two HLA homologs, adjusted to ensure each sequencing read is only counted once.

### Histopathological and immunohistochemistry assessment

Tumor cellularity, extent of TILs, CD3, CD4, CD8 in TILs, and Ki67 in tumor cells were assessed by pathologists (FT, TV, and FP). Tumor cellularity was assessed by histologic examination of hematoxylin and eosin (H&E) stained slides based on the percentage of the tumor area. Extent of TIL infiltration was evaluated following the guidelines put forward by the TIL Working group[66]. In brief, TILs were reported for the stromal compartment (=% stromal TILs; i.e. area occupied by mononuclear inflammatory cells over total intratumoral stromal area). IHC staining for CD3 using the clone LN10 was performed on the Leica Bond-3 auto staining system (Leica, Deerfield, IL, ready-to-use [1 ug/mL]), using heat-based antigen retrieval, a high pH buffer solution (AR9640; Leica, Bond Epitope Retrieval Solution 2), and a polymer detection system (DS9800; Leica, Bond Polymer Refine Detection). CD3 was reported as the area occupied by CD3+ cells over the total intratumoral stromal area[67]. IHC for CD4 and CD8 was performed using the clones SP35 and SP57, respectively, both manufactured by Ventana (Tucson, AZ, ready-to-use). The evaluation was carried out according to the percentage of cytoplasmic positivity expression. Absolute counting of positively stained cells was performed as evaluation criteria to assess IHC for CD4 and CD8 markers[68]. Staining was blinded assessed by an experienced breast cancer pathologist (FT). PD-L1 expression was assessed by IHC using Ventana SP142 assay (dilution 1:50; Spring Bioscience, USA). Immune cells infiltrating tumors with positive staining rate ≥1% were classified as PD-L1-positive. IHC for Ki67 (ready-to-use, Ventana, anti-Ki-67 (30-9)) was performed automatically using an IHC autostainer (BenchMark® XT, Ventana Medical Systems, Inc.). The Ki67 score was calculated obtaining the percentage of positive tumor cells among the total number of tumor cells in each tissue section[69].

### Multiplexed proteomics assessment

For the index case, 35 specimens were subject to multiplex Proximity Extension Assay (PEA) using an immuno-oncology targeted protein panel (Olink Proteomics, Uppsala, Sweden)[70,71]. This comprised 16 body fluids: 14 on-treatment samples (n = 11 serum, n = 3 plasma); 2 postmortem samples (n = 1 ascitic fluid, n = 1 pleural fluid), and 19 protein lysates derived from multiregion metastases taken at the research autopsy procedure. Briefly, 92 oligonucleotide-labeled antibodies bind to the targeted protein and if the two oligonucleotides are in close proximity, a PCR target sequence is formed by a proximity-dependent DNA polymerization event and the resulting sequence is subsequently detected and quantified using real-time PCR. The immunoassay was performed using 1 μL of each sample, which were randomized to avoid batch effects. Detection and quantification were performed by real-time PCR. Each PEA measurement has a specified lower detection limit (LOD) calculated based on negative controls that are included in each run and measurements below this limit were removed from further analysis. Multiplex data were reported in NPX (Normalized Protein Expression) levels, which are Ct values normalized by the subtraction of values for extension control, as well as an interplate control. The scale is then shifted using a run-time specific correction factor (normal background level). The relative quantification of proteins was expressed as NPX (Normalized Protein Expression), an arbitrary unit on log2 scale.

### Statistics

Statistical analyses were performed using R v.3.2 and higher. Before applying the correlation test, we analyzed whether the samples exhibited a normal or quasi-normal distribution based on a quantile-quantile plot and supported by the Shapiro–Wilk test, the absence of outliers, and the presence of a linear relationship between them. If all three assumptions were met, we applied the Pearson correlation test; otherwise, we applied the Spearman correlation test. This approach was consistent for all correlation analyses we conducted.

### Reporting summary

Further information on research design is available in the Nature Portfolio Reporting Summary linked to this article.

## Data availability

The data generated in this study (whole-exome sequencing and RNA-seq of tumor specimens, plus single-cell RNA-seq and single-cell TCRseq of peripheral blood T cells) have been deposited in the European Genome-Phenome Archive (EGA), under restricted access to protect patient information, under the accession code EGAS00001004956. Access will be granted by application to the Data Access Committee (DAC, EGAC50000000074) with responses addressed within 14 working days. Access will be granted for appropriate use for researchers and will be governed by the provisions laid out in the associated informed consent for each cohort or collection, and the terms contained in the Data Access Agreement. Further DNA and RNA-seq data used in this study are publicly available in the NCBI's genotypes and phenotypes database (dbGaP) under accession number phs000676.v1.p1[10] and the EGA database under accession code EGAS00001002703[11]. The remaining data are available within the Article, Supplementary Information or Source Data file. Source data are provided with this paper.

## Code availability

The computational scripts to process the data and plot figures are available here https://doi.org/10.5281/zenodo.10359740[72].

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

## Acknowledgements

The authors express gratitude to the index patient and her family for their participation in this study, and the teams at IrsiCaixa, Rosell Institute of Oncology, Vall d´Hebron Institute of Oncology, the Cellex foundation and NanoString Technologies. "Ayuda Merck de Investigación 2020 in Immuno-Oncology" provided partial support to L.D.M.A. This work was partially supported by internal grants of IrsiCaixa and Vall d´Hebron Institute of Oncology. Research collaboration with NanoString Technologies partially supported the NCounter experiments. No grant numbers apply.   Research reported in this publication was funded in part by a National Institutes of Health (NIH)/National Cancer Institute (NCI) Cancer Center Support Grant (P30 CA008748; to MSK). J.S.R.-F. and F.P. were funded in part by the NIH/NCI P50 CA247749 01 grant; J.S.R.-F. and B.W. by a Breast Cancer Research Foundation grant; J.S.R.-F. by a Susan G Komen leadership grant; and B.W. by a Cycle for Survival grant.The authors thank Nicholas McGranahan and Clare Puttick for scientific discussions regarding measuring and quantifying HLA disruption.

## Author contributions

L.D.M.A. designed the study and wrote the manuscript with the assistance of the co-authors. Data were collected by M.G.C., R.R., S.R. Laboratory work was performed by J.B.H., C.A.S., S.G.R., S.C., S.C., B.W. A.J. Warm autopsy performed by F.T. Pathology analysis performed by F.T., F.P., T.V., A.J. Statistical analysis was performed by C.A.S., J.H.B., M.C., D.T. Computational pipelines ran by J.B.H., C.A.S., J.L.T., M.C., C.P., N.M., D.T., D.P., A.M., AG. Data were analyzed and interpreted by J.B.H., S.G.R., C.A.S., D.P., N.I., B.W., J.S.R.-F., D.T., H.H., L.D.M.A. Co-supervision of single T cell studies done by H.H., L.D.M.A., and C.B. All authors have read and approved the final version of the manuscript.

## Competing interests

L.D.M.A. has received honoraria for participation in a speaker's bureau/consultancy from Roche and reports research collaboration and support from NanoString Technologies, Education grant: BMS, Lilly. L.D.M.A. is currently employed by BioNTech SE. H.H. is co-founder and shareholder of Omniscope, SAB member of Nanostring and MiRXES, and consultant to Moderna and Singularity. C.A.S. is a scientific associate of Dataomics Biotech. J.B.H. is currently affiliated with the Marie-Josée and Henry R. Kravis Center for Molecular Oncology, Memorial Sloan Kettering Cancer Center, New York, NY, USA. J.S.R.-F. reports receiving personal/consultancy fees from Goldman Sachs, Bain Capital, REPARE Therapeutics, Saga Diagnostics, MultiplexDX, and Paige.AI, membership of the scientific advisory boards of VolitionRx, REPARE Therapeutics and Paige.AI, membership of the Board of Directors of Grupo Oncoclinicas, and ad hoc membership of the scientific advisory boards of AstraZeneca, Merck, Daiichi Sankyo, Roche Tissue Diagnostics and Personalis, outside

the scope of this study. J.S.R.-F. is currently employed by AstraZeneca. B.W. reports research funding from Repare Therapeutics, outside the scope of the submitted work. The other authors declare no competing interests.

## Additional information

[1]IrsiCaixa, Germans Trias i Pujol University Hospital, Badalona, Spain. [2]Germans Trias i Pujol Research Institute (IGTP), Badalona, Spain. [3]Department of Pathology and Laboratory Medicine, Memorial Sloan Kettering Cancer Center, New York, NY, USA. [4]Centro Nacional de Análisis Genómico (CNAG), Barcelona, Spain. [5]Josep Carreras Leukemia Research Institute, Barcelona, Spain. [6]Dexeus Institute of Oncology, Quironsalud Group, Barcelona, Spain. [7]Vall d'Hebron Institute of Oncology (VHIO), Vall d'Hebron University Hospital, Barcelona, Spain. [8]Omniscope, Barcelona, Spain. [9]Cancer Research UK Cambridge Institute, Robinson Way, Cambridge, UK. [10]ICREA, Passeig de Lluís Companys, 23, Barcelona, Spain. [11]Universitat de Vic-Universitat Central de Catalunya, Catalunya, Spain. [12]Department of Gynecology and Obstetrics - Breast Disease Division and Laboratory for Translational Data Science, Ribeirao Preto Medical School, University of Sao Paulo, Ribeirao Preto, Brazil. [13]Advanced Research Center in Medicine (CEPAM), Union of the Colleges of the Great Lakes (UNILAGO), São José do Rio Preto, Brazil. [14]These authors contributed equally: Juan Blanco-Heredia, Carla Anjos Souza, Juan L. Trincado, Daniel Guimarães Tiezzi. ✉e-mail: ldmarruda@gmail.com

