## [Peer Review File · Nature Communications]

Converging and evolving immuno-genomic routes towards immune escape in breast cancerEditorial Note: This manuscript has been previously reviewed at another journal that is not operating a transparent peer review scheme. This document only contains reviewer comments and rebuttal letters for versions considered at Nature Communications.

Reviewers' Comments:

Reviewer #2:

Remarks to the Author:

Authors put together a comprehensive response to my previous comments. I believe that they have sufficiently addressed my prior concerns. Their revised manuscript is now substantially improved and more understandable.

Other reviewers raised additional important concerns.

Reviewer #4:

Remarks to the Author:

The authors have by enlarge addressed R1's concerns and the manuscript is now much clearer. However there remains some outstanding points.

Overall, it is confusing what has been performed on the index samples and on the 11 independent samples. I would suggest making this clearer when the analysis is presented in the text.

Major point 2: Although the authors have added additional details to explain the molecular clock analysis, this is still quite confusing. For instance, Figure 5c- what is the x axis here and what exactly is this depicting? It would be useful to include a figure showing a predicted e.g. phylogenetic tree for the different metastatic lesions from this molecular clock analysis.

Major point 4:

The authors have added in additional analyses as suggested by R1 (Figure 2b). However, this new figure is illegible and difficult to interpret. Can the authors present these data in a better way?

Major point 7: Authors have used confusingly 'r' and 'rho' for correlations. They should specify if its Pearson's r and Spearman's rho? and also why interchangeably use parametric and non-parametric statistics?

Major point 9: Authors have performed new heterogeneity analysis, but these new data remain disconnected and lack contextualisation as to how these measures confirm heterogeneity over time. Can this be simplified in some way?

Reviewer #5:

Remarks to the Author:

With regard to responses provided to Reviewer #3, the authors have satisfied the concerns by the original reviewer.

For consideration:

The addition of the lengthy clinical history of the index patient to the main text may make Supp Table 1 redundant. It may also be more concise to structure the legend of Figure 1a with more details and shorten the main body text. You might consider converting larger numbers in the text to months or years. For example, 2033 days to "over 5 years of clinical follow-up". Consider "expired" or "died" rather than "passed away"

Figure 1b could be more informative. As currently configured, it is binary (yes/no) for data from the primary and merely counts the number of metastatic deposits sampled. It could combine the information in Supp Table 2 with codes for sites of metastases. For example, the circles for A1 could be AD, LN, LU, LI, and SP for Adrenal, Lymph Node, Lung, Liver and Spine. Readers know at a glance what type of tissue you're looking at. As it stands now, it provides little additional information.

REVIEWER COMMENTS

Reviewer #2 (Remarks to the Author):

Authors put together a comprehensive response to my previous comments. I believe that they have sufficiently addressed my prior concerns. Their revised manuscript is now substantially improved and more understandable.

Response: We thank Reviewer #2 for the kind comment.

Other reviewers raised additional important concerns.

Reviewer #4, assessed responses to previous Reviewer's #1 comments (Remarks to the Author):

The authors have by enlarge addressed R1's concerns and the manuscript is now much clearer. However there remains some outstanding points.

Overall, it is confusing what has been performed on the index samples and on the 11 independent samples. I would suggest making this clearer when the analysis is presented in the text.

Response: One particular area of concern is the clarity regarding which analyses have been conducted on the index samples and which have been performed on the 11 independent TNBC patients. To enhance the manuscript's clarity, we specified and differentiated between the analyses conducted on the index TNBC patient and those performed on the 11 independent TNBC patients when presenting the results in section "Clinical and sample characteristics of TNBC patients" of Results, pages 2-3 and across the text.

Major point 2: Although the authors have added additional details to explain the molecular clock analysis, this is still quite confusing. For instance, Figure 5c- what is the x axis here and what exactly is this depicting? It would be useful to include a figure showing a predicted e.g. phylogenetic tree for the different metastatic lesions from this molecular clock analysis.

Response:

We acknowledge this comment of the reviewer and have made the suggested modifications. We have generated and included in figure 5c a predicted hierarchical clustering analysis for the different early tumors and metastatic lesions from our molecular clock analysis. That helped us to map the metastases into different evolutionary branches and to better understand that similar chronological or molecular time origins can exhibit varying genomic and immune environment landscapes across parallel multiregion metastases.

In Figure 5c, the y-axis in the new representation is depicting **Mutation Time** as defined by T1.histBeta. The T1.histBeta value (found in Supplementary Table 11) was determined by identifying the mode of the initial copy number gains in each sample. This value served as a marker for each sample's molecular time, encompassing both Mutation and Neoantigen time, and facilitated the ordering of parallel multiregion metastases.

We made edits to the legends, and they now read as follows:

c, Molecular clock hierarchical clustering analysis depicting **Mutation Time** (y-axis, defined by T1.histBeta) for sequential P0 (primary), M0-day 373, M1-day 799, M3-day 1,687 chest wall tumors, and 20 parallel multiregion metastases. Histograms illustrate the distribution of event timing within these samples, categorized as early (lowest quartile, Q25), intermediate (inter, Q25-75), or late (highest quartile, Q75) based on the criteria detailed in the methods section.

The text can be read as follows:

“The Mutation Time (T1.histBeta) defined each sample’s molecular time and was used to molecularly order sequential and parallel multiregion metastases. As expected, the primary breast cancer-P0, and temporally earlier chest wall metastases (M0-day 373, M1 day-799 and M3 day-1,687) had an early Mutation Time, reinforcing the robustness of our analyses (Fig. 5c). A Molecular Time hierarchical clustering analysis revealed two main branches (Fig. 5c). The first branch was defined by samples with early Mutation and Chronological Time (i.e., P0, M0-1). In contrast, the second branch represented a later origin, including metastases classified as late with high Mutation Time. The Mutation Time did not follow a linear, contiguous order per organ or multi-tissue site, potentially skipping contiguous areas of the lung, diaphragm, pericardium, mediastinal lymph nodes, or chest wall. There was an enrichment of subclonal mutations in metastases classified as “late” as compared to “early” (p-value=0.02; **Fig. 5d**). These observations underscore the complexity of the evolutionary timeline and the genomic diversity inherent in this TNBC index patient. Similar chronological or molecular time origins exhibited varying genomic landscapes, resulting in increased heterogeneity across tumors and metastases.

In Methods we added as follows:

“The dendrogram was constructed using the 'clustermap' function from the Seaborn library in Python, which executes hierarchical clustering. In this analysis, timed copy number variants derived from MutationTimeR served as the input dataset and default function parameters.”

Major point 4:

The authors have added in additional analyses as suggested by R1 (Figure 2b). However, this new figure is illegible and difficult to interpret. Can the authors present these data in a better way?

Response: We thank the reviewer for this comment. We have generated a new figure to substitute **Figure 2b**. The new figure shows the TCR β CDR3 repertoire joined network of the on-treatment sequential metastases (yellow), parallel multiregion metastases (red) and present in both tumor sources (purple). We have added an insert below the network figure panel highlighting amino acid sequences from a subnetwork of metastases specific to parallel multiregions. Each node's size corresponds to the number of samples where the sequence has

been detected. Edges were formed between nodes only when the edit distance between the two CDR3 sequences equaled 1.

We modified the legends of figure 2b and it can be read as follows:

“TCR β CDR3 repertoire joined network to elucidate subnetworks private to on-treatment sequential chest wall metastases (yellow) or to parallel multiregion metastases (red) or shared to both sets of metastases (purple). Insert on the bottom shows amino acid sequences from a parallel multiregion metastases-specific subnetwork. Each node's size corresponds to the number of samples where the sequence has been detected. Edges were formed between nodes only when the edit distance between the two CDR3 sequences equaled 1.”

Major point 7: Authors have used confusingly 'r' and 'rho' for correlations. They should specify if its Pearson's r and Spearman's rho? and also why interchangeably use parametric and non-parametric statistics?

Response: Regarding the comment about the use of 'r' and 'rho' for correlations, we have explained our rationale for using both types of tests based on the specific characteristics of the data and the adherence to underlying assumptions.

Before applying the correlation test, we analyzed whether the samples exhibited a normal or quasi-normal distribution based on a quantile-quantile plot and supported by the Shapiro Wilk test, the absence of outliers, and the presence of a linear relationship between them. If all three assumptions were met, we applied the Pearson correlation test; otherwise, we applied the Spearman correlation test. This approach was consistent for all correlation analyses we conducted.

Major point 9: Authors have performed new heterogeneity analysis, but these new data remain disconnected and lack contextualisation as to how these measures confirm heterogeneity over time. Can this be simplified in some way?

Response:

We have included figure 5c showing a predicted hierarchical clustering analysis for the different early tumors and metastatic lesions of the TNBC patient from our molecular clock analysis. That helped us to map the metastases into different evolutionary branches and to better understand that similar chronological or molecular time origins can exhibit varying genomic and immune environment landscapes across parallel multiregion metastases.

We concluded that: “These observations [based on the predicted hierarchical clustering analysis for the different early tumors and metastatic lesions from our molecular clock] underscore the complexity of the evolutionary timeline and the genomic diversity inherent in this TNBC index patient. Similar chronological or molecular time origins exhibited varying genomic landscapes, resulting in increased heterogeneity across tumors and metastases.”

We conclude the section with: “Metastases with similar molecular time origins exhibited varying genomic and immune microenvironment landscapes. This intricate relationship underscores the need for precise molecular profiling and tailored treatments to address the diverse aspects of cancer evolution.”

Reviewer #5, assessed responses to previous Reviewer's #3 comments (Remarks to the Author):

With regard to responses provided to Reviewer #3, the authors have satisfied the concerns by the original reviewer.

For consideration:

The addition of the lengthy clinical history of the index patient to the main text may make Supp Table 1 redundant. It may also be more concise to structure the legend of Figure 1a with more details and shorten the main body text. You might consider converting larger numbers in the text to months or years. For example, 2033 days to "over 5 years of clinical follow-up". Consider "expired" or "died" rather than "passed away".

Response:

We agree with the reviewer to make the text more concise. As suggested, we have structured the legend of Figure 1a with more details of the clinical history of the index patient and shorten the main body text.

The body can be read as follows:

“An index patient with TNBC was closely monitored from diagnosis through metastatic progression for 2,033 days until she expired. Data included samples from various stages of the disease, such as a primary tumor specimen, 15 sequential chest wall skin biopsies during treatment, 18 blood samples, 2 body fluids (ascitic and pleural), and 20 parallel metastases from multiple regions.

The patient, a 49-year-old woman, initially had stage III TNBC with a 3.5cm breast mass and lymph node involvement. She underwent various treatments, achieving a complete response after neoadjuvant chemotherapy. Recurrences occurred over time, and immunotherapy with atezolizumab showed a fast disease progression but later contributed to a prolonged response to cisplatin and gemcitabine. The patient underwent several treatment regimens for metastatic disease, including cytotoxic and immunotherapies, and ultimately expired after 2,033 days of clinical follow-up (**Fig. 1, Supplementary Table 1 and 2**).”

The legends of Fig 1 can be read as follows:

“Figure 1. Study schematics.

a, Schematics of the study and anatomical map of biospecimen collection for sequential and parallel multiregion analyses of the index TNBC patient.

The index patient presented here was a 49-year-old woman diagnosed with a stage III TNBC (T2N3, estrogen receptor (ER), progesterone receptor (PR) and HER2 0+ negative, Ki67 60%, grade 3) with a 3.5cm right breast cancer mass and lymph node involvement, who underwent multiple systemic therapies due to recurrences and metastatic progression over 2,033 days of clinical follow-up. She underwent neoadjuvant chemotherapy with anthracycline and taxane achieving a pathological complete response after mastectomy. The patient presented multiple clinical recurrences at the chest wall from day-373, and achieved complete response with cisplatin-based therapy, surgery and local radiotherapy. A second chest wall recurrence occurred around day-666 with partial response to bevacizumab-based therapy. Immunotherapy employing anti-PD-L1 monoclonal antibody atezolizumab was administered on day-799 after

diagnosis, followed by in a rapid disease progression. However, a long lasting complete response of 22 months was evidenced after a re-challenge of cisplatin and gemcitabine (day-854 to day-1519), after a previous response to the same drug had been 2.2 fold shorter (day-373 to day-666). The anti-PD-L1 administration before the re-challenge with cisplatin, although culminating in a rapid disease progression, could have contributed to the subsequent long-lasting antitumor response to cisplatin, motivating our investigation of immune escape. Subsequently, the patient presented a new progression at the chest wall, and received anti-PD1 (pembrolizumab) and chemotherapy, with stable disease for 4 months. Then, upon chest wall progression, pembrolizumab plus toll-like receptor (TLR) 7 agonist (topical) was administered in the chest wall metastases with a transient local complete response that lasted around 50 days. The patient received other lines of systemic therapy (i.e., palbociclib followed by cyclophosphamide, pegylated liposomal doxorubicin, cisplatin plus gemcitabine, paclitaxel plus bevacizumab, eribulin) (**Supplementary Table 1**) and expired on day 2,033.

Sequential chest wall images illustrate the clinical evolution of a TNBC patient over time. Post-mortem parallel multiregion metastases were synchronous, affecting the same metastasis or metastases affecting different anatomical sites (separated into 2 or 3 sites when indicated) within the index patient as indicated. “

We acknowledge the review suggestion to convert larger numbers in the text to months or years. However, we would like to keep the numbers (e.g., day 2,033) for consistency in the text and figures.

We have altered "passed away" for "expired" on pages 2, 10 and 25.

Figure 1b could be more informative. As currently configured, it is binary (yes/no) for data from the primary and merely counts the number of metastatic deposits sampled. It could combine the information in Supp Table 2 with codes for sites of metastases. For example, the circles for A1 could be AD, LN, LU, LI, and SP for Adrenal, Lymph Node, Lung, Liver and Spine. Readers know at a glance what type of tissue you're looking at. As it stands now, it provides little additional information.

Response: As recommended for Figure 1b, we have integrated the data from Supplementary Table 2, along with the corresponding codes for metastatic sites. These details have been included within Figure 1b itself and have been updated in the captions for Figure 1b.

The revised descriptions are as follows:

AD, adrenal; BO, bone; BR, brain; BT, breast [metastasis]; CH, chest; KI, kidney; LN, lymph node; LU, lung; LI, liver; ME, meninges; PB, primary breast; PE, pleura; SP, Spine; ST, soft tissue; SK, skin.

Reviewers' Comments:

Reviewer #4:

Remarks to the Author:

The authors have sufficiently addressed the remaining points, and made the manuscript clearer.

Reviewer #5:

Remarks to the Author:

The authors have addressed my concerns. The manuscript has been strengthened and clarified.

REVIEWERS' COMMENTS

Reviewer #4 (Remarks to the Author):

The authors have sufficiently addressed the remaining points, and made the manuscript clearer.

Reviewer #5 (Remarks to the Author):

The authors have addressed my concerns. The manuscript has been strengthened and clarified.

Reply: We thank the reviewers for these positive comments.